# Contaminated Online Convex Optimization

**Tomoya Kamijima**                                          *kamijima-tomoya101@g.ecc.u-tokyo.ac.jp*
*Department of Mathematical Informatics*
*The University of Tokyo*
**Shinji Ito**                                                        *shinji@mist.i.u-tokyo.ac.jp*
*Department of Mathematical Informatics*
*The University of Tokyo*
*NEC Corporation (affiliation upon submission)*
*RIKEN AIP*

**Reviewed on OpenReview:** *https://openreview.net/forum?id=QdGtwjDgub*

## Abstract

In online convex optimization, some efficient algorithms have been designed for each of the individual classes of objective functions, e.g., convex, strongly convex, and exp-concave. However, existing regret analyses, including those of universal algorithms, are limited to cases in which the objective functions in all rounds belong to the same class and cannot be applied to cases in which the property of objective functions may change in each time step. This paper introduces a novel approach to address such cases, proposing a new regime we term as *contaminated* online convex optimization. For the contaminated case, we demonstrate that the regret is lower bounded by $\Omega(\log T + \sqrt{k})$. Here, $k$ signifies the level of contamination in the objective functions. We also demonstrate that the regret is bounded by $O(\log T + \sqrt{k \log T})$ when universal algorithms are used. When our proposed algorithms with additional information are employed, the regret is bounded by $O(\log T + \sqrt{k})$, which matches the lower bound. These are intermediate bounds between a convex case and a strongly convex or exp-concave case.

## 1 Introduction

Online convex optimization (OCO) is an optimization framework in which convex objective function changes for each time step $t \in \{1, 2, \ldots, T\}$. OCO has a lot of applications such as prediction from expert advice (Littlestone & Warmuth, 1994; Arora et al., 2012), spam filtering (Hazan, 2016), shortest paths (Awerbuch & Kleinberg, 2004), portfolio selection (Cover, 1991; Hazan et al., 2006), and recommendation systems (Hazan & Kale, 2012). The performance of the OCO algorithm is compared by regret (defined in Section 3). As shown in Table 1, it is already known that sublinear regret can be achieved for each function class, such as convex, strongly convex, and exp-concave, and the bound depends on the function class. In addition, these upper bounds coincide with lower bounds, so these are optimal. However, these optimal algorithms are applicable to one specific function class. Therefore, we need prior knowledge about the function class to which the objective functions belong.

To solve this problem, many universal algorithms that work well for multiple function classes by one algorithm have been proposed (Hazan et al., 2007; Van Erven & Koolen, 2016; Wang et al., 2020; Zhang et al., 2022; Yan et al., 2024). For example, the MetaGrad algorithm proposed by Van Erven & Koolen (2016) achieves an $O(\sqrt{T})$-regret for any sequence of convex objective functions and an $O(\log T)$-regret if all the objective functions are exp-concave. Universal algorithms are useful in that they can be used without prior knowledge about the objective functions. Some universal algorithms are introduced in Appendix A.3.

A notable limitation of the previous regret analyses about universal algorithms is that they apply only to cases where all the objective functions $f_1, f_2, \ldots, f_T$ belong to the same function class. Therefore, for example, if some objective functions in a limited number of rounds are not strongly convex and if the other

Table 1: Comparison of regret bounds. The parameter $d$ is the dimension of the decision set.

| FUNCTION CLASS | UPPER BOUNDS | LOWER BOUNDS |
|---|---|---|
| Convex | $O(\sqrt{T})$ (Zinkevich, 2003) | $\Omega(\sqrt{T})$ (Abernethy et al., 2008) |
| $\alpha$-exp-concave | $O((d/\alpha)\log T)$ (Hazan et al., 2006) | $\Omega((d/\alpha)\log T)$ (Ordentlich & Cover, 1998) |
| $k$-contaminated $\alpha$-exp-concave | $O((d/\alpha)\log T + \sqrt{kd\log T})$ **(This work, Corollary 5.5)** | $\Omega((d/\alpha)\log T + \sqrt{k})$ **(This work, Corollary 4.8)** |
| $k$-contaminated $\alpha$-exp-concave (with additional information) | $O((d/\alpha)\log T + \sqrt{k})$ **(This work, Theorem 6.2)** | $\Omega((d/\alpha)\log T + \sqrt{k})$ **(This work, Corollary 4.8)** |
| $\lambda$-strongly convex | $O((1/\lambda)\log T)$ (Hazan et al., 2006) | $\Omega((1/\lambda)\log T)$ (Abernethy et al., 2008) |
| $k$-contaminated $\lambda$-strongly convex | $O((1/\lambda)\log T + \sqrt{k\log T})$ **(This work, Corollary 5.7)** | $\Omega((1/\lambda)\log T + \sqrt{k})$ **(This work, Corollary 4.9)** |
| $k$-contaminated $\lambda$-strongly convex (with additional information) | $O((1/\lambda)\log T + \sqrt{k})$ **(This work, Theorem 6.2)** | $\Omega((1/\lambda)\log T + \sqrt{k})$ **(This work, Corollary 4.9)** |

objective functions are strongly convex, regret bounds for strongly convex functions in previous studies are not always valid. This study aims to remove this limitation.

## 1.1 Our Contribution

In this study, we consider the situation in which the function class of the objective $f_t$ may change in each time step $t$. We call this situation *contaminated* OCO. More specifically, in $k$-contaminated OCO with a function class $\mathcal{F}$, we suppose that the objective function $f_t$ does not necessarily belong to the desired function class $\mathcal{F}$ (e.g., exp-concave or strongly convex functions) in $k$ rounds out of the total $T$ rounds. Section 3 introduces its formal definition and examples. This class of OCO problems can be interpreted as an intermediate setting between general OCO problems and restricted OCO problems with $\mathcal{F}$ ($\mathcal{F}$-OCOs). Intuitively, the parameter $k \in [0, T]$ represents the magnitude of the impurity in the sequence of the objective functions, and measures how close the problems are to $\mathcal{F}$-OCOs; $k = 0$ and $k = T$ respectively correspond to $\mathcal{F}$-OCO and general OCO.

The contribution of this study can be summarized as follows: (i) We introduce contaminated OCO, which captures the situations in which the class of the objective functions may change over different rounds. (ii) We find that the Online Newton Step, one of the optimal algorithms for exp-concave functions, does not always work well in contaminated OCO, as discussed in Section 4.1. (iii) We present regret lower bounds for contaminated OCO in Section 4.2. (iv) We show that some existing universal algorithms achieve better regret bounds than ONS for contaminated OCO, which details are given in Section 5. (v) We propose an algorithm that attains the optimal regret bounds under the additional assumption that information of the class of the previous objective function is accessible in Section 6.

Regret bounds of contaminated cases compared to existing bounds are shown in Table 1. The new upper bounds contain bounds in existing studies for exp-concave functions and strongly convex functions as a particular case ($k = 0$). Additionally, the new lower bounds generalize bounds in existing studies for convex functions, exp-concave functions, and strongly convex functions. In cases where only gradient information is available, there is a multiplicative gap of $O(\sqrt{\log T})$ between the second terms of the upper bounds and the lower bounds. This gap is eliminated when the information of the class of the previous objective function is available.

To prove regret lower bounds in Table 1, we construct distributions of problem instances of contaminated OCO for which any algorithm suffers a certain amount of regret in expected values. Such distributions are constructed by combining suitably designed problem instances of $\mathcal{F}$-OCO and general OCO.

To derive novel regret upper bounds without additional information in Table 1, we exploit regret upper bounds expressed using some problem-dependent values such as a measure of variance (Van Erven & Koolen, 2016). By combining such regret upper bounds and inequalities derived from the definition of $k$-contaminated OCO, we obtain regret upper bounds, including the regret itself, which can be interpreted as quadratic inequalities in regret. Solving these inequalities leads to regret upper bounds in Table 1.

We develop algorithms that can achieve optimal regret upper bounds, taking into account the function class information of the previous function. To accomplish this, we modified two existing OCO algorithms: the Online Newton Step (ONS), as introduced by Hazan et al. (2006), and the Online Gradient Descent (OGD), presented by Zinkevich (2003). The modification is changing the update process depending on the function class of the last revealed objective function.

## 2 Related Work

In the context of online learning, *adaptive* algorithms (Orabona, 2019) have been extensively studied due to their practical usefulness. These algorithms work well by automatically exploiting the intrinsic properties of the sequence of objective functions and do not require parameter tuning based on prior knowledge of the objective function. For example, AdaGrad (McMahan & Streeter, 2010; Duchi et al., 2011) is probably one of the best-known adaptive algorithms, which automatically adapts to the magnitude of the gradients. Studies on *universal* algorithms (Hazan et al., 2007; Van Erven & Koolen, 2016; Wang et al., 2020; Zhang et al., 2022; Yan et al., 2024), which work well for several different function classes, can also be positioned within these research trends. Our study shows that some of these universal algorithms have further adaptability, i.e., nearly tight regret bounds for contaminated settings.

van Erven et al. (2021) has explored a similar setting to ours, focusing on robustness to *outliers*. They regard rounds with larger gradient norms than some threshold as outliers and denote the number of outliers as $k$, whose definition differs from ours. They have defined regret only for rounds that are not outliers, terming it *robust regret*, and have shown that the additional $O(k)$ term is inevitable in bounds on robust regret.

Studies on best-of-both-worlds (BOBW) bandit algorithms (Bubeck & Slivkins, 2012) and on stochastic bandits with adversarial corruptions (Lykouris et al., 2018; Gupta et al., 2019) are also related to our study. BOBW algorithms are designed to achieve (nearly) optimal performance both for stochastic and adversarial environments, e.g., $O(\log T)$-regret for stochastic and $O(\sqrt{T})$-regret for adversarial environments, respectively. Stochastic bandits with adversarial corruptions are problems for intermediate environments between stochastic and adversarial ones, in which the magnitude of adversarial components is measured by means of the *corruption level* parameter $C \geq 0$. A BOBW algorithm by Bubeck & Slivkins (2012) has shown to have a regret bound of $O(\log T + \sqrt{C \log T})$ as well for stochastic environments with adversarial corruptions, which is also nearly tight (Ito, 2021). In the proof of such an upper bound, an approach referred to as the *self-bounding technique* (Gaillard et al., 2014; Wei & Luo, 2018) is used, which leads to improved guarantees via some regret upper bounds that include the regret itself. Similar proof techniques are used in our study as well.

## 3 Problem Setting

In this section, we explain the problem setting we consider. Throughout this paper, we assume functions $f_1, f_2, \ldots, f_T$ are differentiable and convex.

### 3.1 OCO Framework and Assumptions

First of all, the mathematical formulation of OCO is as follows. At each time step $t \in [T] (\coloneqq \{1, 2, \ldots, T\})$, a convex nonempty set $\mathcal{X} \subset \mathbb{R}^d$ and convex objective functions $f_1, f_2, \ldots, f_{t-1} \colon \mathcal{X} \to \mathbb{R}$ are known and $f_t$

is not known. A learner chooses an action $\boldsymbol{x}_t \in \mathcal{X}$ and incurs a loss $f_t(\boldsymbol{x}_t)$ after the choice. Since $f_t$ is unknown when choosing $\boldsymbol{x}_t$, it is impossible to minimize the cumulative loss $\sum_{t=1}^{T} f_t(\boldsymbol{x}_t)$ for all sequences of $f_t$. Instead, the goal of OCO is to minimize regret:

$$R_T := \sum_{t=1}^{T} f_t(\boldsymbol{x}_t) - \min_{\boldsymbol{x} \in \mathcal{X}} \sum_{t=1}^{T} f_t(\boldsymbol{x}).$$

Regret is the difference between the cumulative loss of the learner and that of the best choice in hindsight. The regret can be logarithmic if the objective functions are $\lambda$-strongly convex, i.e., $f(\boldsymbol{y}) \geq f(\boldsymbol{x}) + \langle \nabla f(\boldsymbol{x}), \boldsymbol{y} - \boldsymbol{x} \rangle + \frac{\lambda}{2} \|\boldsymbol{x} - \boldsymbol{y}\|^2$ for all $\boldsymbol{x}, \boldsymbol{y} \in \mathcal{X}$, or $\alpha$-exp-concave, i.e., $\exp(-\alpha f(\boldsymbol{x}))$ is concave on $\mathcal{X}$.

*Remark* 3.1. The type of information about $f_t$ that needs to be accessed varies depending on the algorithm. Universal algorithms only utilize gradient information, while the algorithm presented in Section 6 requires additional information besides the gradient, such as strong convexity and exp-concavity. The lower bounds discussed in Section 4.2 are applicable to arbitrary algorithms with complete access to full information about the objective functions.

Next, we introduce the following two assumptions. These assumptions are very standard in OCO and frequently used in regret analysis. We assume them throughout this paper without mentioning them.

**Assumption 3.2.** There exists a constant $D > 0$ such that $\|\boldsymbol{x} - \boldsymbol{y}\| \leq D$ holds for all $\boldsymbol{x}, \boldsymbol{y} \in \mathcal{X}$.

**Assumption 3.3.** There exists a constant $G > 0$ such that $\|\nabla f_t(\boldsymbol{x})\| \leq G$ holds for all $\boldsymbol{x} \in \mathcal{X}$ and $t \in [T]$.

These assumptions are important, not only because we can bound $\|\boldsymbol{x} - \boldsymbol{y}\|$ and $\|\nabla f_t(\boldsymbol{x})\|$, but also because we can use the following two lemmas:

**Lemma 3.4.** *(Hazan, 2016) Let $f \colon \mathcal{X} \to \mathbb{R}$ be an $\alpha$-exp-concave function. Assume that there exist constants $D, G > 0$ such that $\|\boldsymbol{x} - \boldsymbol{y}\| \leq D$ and $\|\nabla f(\boldsymbol{x})\| \leq G$ hold for all $\boldsymbol{x}, \boldsymbol{y} \in \mathcal{X}$. The following holds for all $\gamma \leq (1/2)\min\{1/(GD), \alpha\}$ and all $\boldsymbol{x}, \boldsymbol{y} \in \mathcal{X}$:*

$$f(\boldsymbol{x}) \geq f(\boldsymbol{y}) + \langle \nabla f(\boldsymbol{y}), \boldsymbol{x} - \boldsymbol{y} \rangle + \frac{\gamma}{2}(\langle \nabla f(\boldsymbol{y}), \boldsymbol{x} - \boldsymbol{y} \rangle)^2.$$

**Lemma 3.5.** *(Hazan, 2016) If $f \colon \mathcal{X} \to \mathbb{R}$ is a twice differentiable $\lambda$-strongly convex function satisfying $\|\nabla f(\boldsymbol{x})\| \leq G$ for all $\boldsymbol{x} \in \mathcal{X}$, then it is $\lambda/G^2$-exp-concave.*

Lemma 3.5 means that exp-concavity is a milder condition than strong convexity, combining with the fact that $-\log\langle \boldsymbol{a}, \boldsymbol{x} \rangle$ is not strongly convex but 1-exp-concave.

## 3.2 Contaminated Case

In this subsection, we define *contaminated* OCO and introduce examples that belong to this problem class. The definition is below.

**Definition 3.6.** For some function class $\mathcal{F}$, a sequence of convex functions $(f_1, f_2, \ldots, f_T)$ belongs to $k$-*contaminated $\mathcal{F}$ if there exists a set of indices $I \subset [T]$ such that $|I| = k$ and $f_t \in \mathcal{F}$ holds for all $t \in [T] \backslash I$.*

For example, if functions other than $k$ functions of them are $\alpha$-exp-concave, we call the functions $k$-contaminated $\alpha$-exp-concave. And especially for OCO problems, if the objective functions are contaminated, we call them contaminated OCO.

The following two examples are functions whose function class varies with time step. These examples motivate this study.

*Example* 3.7. (Online least mean square regressions) Consider the situation where a batch of input-output data $(\boldsymbol{a}_{t,i}, b_{t,i}) \in \mathbb{R}^d \times \mathbb{R}$ ($i \in \{1, 2, \ldots, n\}$) is given in each round $t$ and we want to estimate $\boldsymbol{x}$ which enable to predict $b \approx \langle \boldsymbol{a}, \boldsymbol{x} \rangle$. This can be regarded as an OCO problem whose objective functions are $f_t(\boldsymbol{x}) := (1/n)\sum_{i=1}^{n}(\langle \boldsymbol{a}_{t,i}, \boldsymbol{x} \rangle - b_{t,i})^2$. These functions are $\lambda_t$-strongly convex, where $\lambda_t$ is the minimum eigenvalue of the matrix $(2/n)\sum_{i=1}^{n} \boldsymbol{a}_{t,i}\boldsymbol{a}_{t,i}^{\top}$. Let $k(\lambda) := |\{t \in [T] \mid \lambda_t < \lambda\}|$ for any $\lambda > 0$. Then $(f_1, f_2, \ldots, f_T)$ is $k(\lambda)$-contaminated $\lambda$-strongly convex.

*Example* 3.8. (Online classification by using logistic regression) Consider the online classification problem. A batch of input-output data $(\boldsymbol{a}_{t,i}, b_{t,i}) \in \mathbb{R}^d \times \{\pm 1\}$ ($i \in \{1, 2, \ldots, n\}$) is given in each round $t$ and we want to estimate $\boldsymbol{x}$ which enable to predict $b = \mathrm{sgn}(\langle \boldsymbol{a}, \boldsymbol{x} \rangle)$. Suppose that the objective functions are given by $f_t(\boldsymbol{x}) := (1/n) \sum_{i=1}^{n} \log(1 + \exp(-b_{t,i} \langle \boldsymbol{a}_{t,i}, \boldsymbol{x} \rangle))$. Exp-concavity of $f_t(\boldsymbol{x})$ on $\{\boldsymbol{x} \in \mathbb{R}^d \mid \|\boldsymbol{x}\| \leq 1\}$ changes with time step. Especially, in the case $\boldsymbol{a}_{t,i} = \boldsymbol{a}_t$, $b_{t,i} = b_t$, $f_t$ is $\exp(-\|\boldsymbol{a}_t\|)$-exp-concave, as proved in Appendix B.1. Let $k(\alpha) := |\{t \in [T] \mid \alpha_t < \alpha\}|$ for any $\alpha > 0$, where $\alpha_t$ is defined so that $f_t$ is $\alpha_t$-exp-concave. Then $(f_1, f_2, \ldots, f_T)$ is $k(\alpha)$-contaminated $\alpha$-exp-concave.

*Remark* 3.9. In the two examples above, constants $\lambda$ and $\alpha$ in the definition of $\lambda$-strong convexity and $\alpha$-exp-concavity can be strictly positive for all time steps. However, since the regret bounds are $O((1/\lambda) \log T)$ and $O((d/\alpha) \log T)$ for $\lambda$-strongly convex functions and $\alpha$-exp-concave functions respectively, if $\lambda$ and $\alpha$ are $O(1/T)$, then the regret bounds become trivial. Analyses in this paper give a nontrivial regret bound to such a case.

## 4 Regret Lower Bounds

### 4.1 Vulnerability of ONS

This subsection explains how Online Newton Step (ONS) works for contaminated exp-concave functions. ONS is an algorithm for online exp-concave learning. Details of ONS are in Appendix A.2. The upper bound is as follows.

**Proposition 4.1.** *If a sequence of objective functions $(f_1, f_2, \ldots, f_T)$ is $k$-contaminated $\alpha$-exp-concave, the regret upper bound of ONS with $\gamma = (1/2) \min\{1/(GD), \alpha\}$ and $\varepsilon = 1/(\gamma^2 D^2)$ is $O((d/\gamma) \log T + k)$.*

This proposition is proved by using the proof for noncontaminated cases by Hazan (2016). A detailed proof is in Appendix B.2. This upper bound seems trivial, but the bound is tight because of the lower bound stated in Corollary 4.6.

Before stating the lower bound, we introduce the following theorem, which is essential in deriving some lower bounds of contaminated cases.

**Theorem 4.2.** *Let $\mathcal{F}$ be an arbitrary function class. Suppose that functions $g_1, g_2$ are the functions such that $\Omega(g_1(T))$ and $\Omega(g_2(T))$ are lower bounds for function class $\mathcal{F}$ and convex functions, respectively, for some OCO algorithm. If a sequence of objective functions belongs to $k$-contaminated $\mathcal{F}$, then regret in the worst case is $\Omega(g_1(T) + g_2(k))$ for the OCO algorithms.*

*Remark* 4.3. In Theorem 4.2, if the lower bounds $\Omega(g_1(T))$ and $\Omega(g_2(T))$ are for all OCO algorithms, then the lower bound $\Omega(g_1(T) + g_2(k))$ is also for all OCO algorithms.

To derive this lower bound, we use the following two instances; one is the instance used to prove lower bound $R_T = \Omega(g_1(T))$ for function class $\mathcal{F}$, and the other is the instance used to prove $R_k = \Omega(g_2(k))$ for convex objective functions. By considering the instance that these instances realize with probability $1/2$, we can construct an instance that satisfies

$$\mathbb{E}[R_T] = \Omega(g_1(T) + g_2(k)),$$

for all OCO algorithms. A detailed proof of this proposition is postponed to Appendix B.3.

Theorem 4.2 implies that, in contaminated cases, we can derive interpolating lower bounds of regret. The lower bound obtained from this theorem is $\Omega(g_1(T))$ if $k \ll T$, and $\Omega(g_2(T))$ if $k = T$. Since the contaminated case can be interpreted as an intermediate regime between restricted $\mathcal{F}$-OCO and general OCO, this result would seen as reasonable. This lower bound applies not only to ONS but also to arbitrary algorithms.

To apply Theorem 4.2 to ONS, we show the following lower bound in the case of convex functions. This lower bound shows that ONS is not suitable for convex objective functions.

**Proposition 4.4.** *For any positive parameters $\gamma$ and $\varepsilon$, ONS incurs $\Omega(T)$ regret in the worst case.*

To prove this proposition, consider the instance as follows:

$$f_t(x) = v_t x, \ x \in \mathcal{X} = [-D/2, D/2], \ x_1 = -G,$$

where

$$v_t = \begin{cases} (-1)^t G & t < t_1, \\ G & t \geq t_1, \ x_{t_1} \geq 0, \\ -G & t \geq t_1, \ x_{t_1} < 0, \end{cases}$$

and $t_1$ is a minimum natural number which satisfies $t_1 \geq (1 + \gamma G^2 D/2)^{-1} T$. Then, we can get

$$R_T \geq \frac{\gamma G^2 D/2}{2(1 + \gamma G^2 D/2)^2} T - \frac{1}{\gamma} \log \left(1 + \frac{G^2}{\varepsilon} T\right) - \frac{2}{\gamma G} - \frac{G^2 D}{2}.$$

A detailed proof of this proposition is postponed to Appendix B.4.

*Remark* 4.5. Corollary 4.4 states the lower bound that holds only for ONS. However, if some better algorithms are used, the lower bound can be improved. Therefore, it is not a contradiction that the general lower bound in Table 1 is better than that of ONS. This is also true for Corollary 4.6, which is about the contaminated case.

The lower bound of $\alpha$-exp-concave functions can be derived as follows. The lower bound of 1-exp-concave functions is $\Omega(d \log T)$ (Ordentlich & Cover, 1998). Here, when divided by $\alpha$, 1-exp-concave functions turn into $\alpha$-exp-concave functions, and regret is also divided by $\alpha$. Hence, the lower bound of $\alpha$-exp-concave functions is $\Omega((d/\alpha) \log T)$.

We get the following from this lower bound for exp-concave functions, Proposition 4.4, and Theorem 4.2.

**Corollary 4.6.** *If a sequence of objective functions $(f_1, f_2, \ldots, f_T)$ is $k$-contaminated $\alpha$-exp-concave, regret in worst case is $\Omega((d/\alpha) \log T + k)$, for ONS.*

This proposition shows that the regret analysis in Proposition 4.1 is strict. While ONS does not work well for contaminated OCO, universal algorithms exhibit more robust performance. In Section 5, we analyze some universal algorithms on this point.

*Remark* 4.7. For the 1-dimension instance above, ONS can also be regarded as OGD (Algorithm 2 in Appendix A.1) with $\Theta(1/t)$ stepsize. This implies that we can show that OGD with $\Theta(1/t)$ stepsize can incur $\Omega(T)$ regret in the worst case. Therefore, for $k$-contaminated strongly convex functions, regret in worst case is $\Omega((1/\lambda) \log T + k)$, for OGD with $\Theta(1/t)$ stepsize.

### 4.2 General Lower Bounds

In this subsection, we present regret lower bounds for arbitrary algorithms.

Using Theorem 4.2, we can get a lower bound of $k$-contaminated exp-concave functions. As mentioned in Section 4.1, regret lower bound of $\alpha$-exp-concave functions is $\Omega((d/\alpha) \log T)$. From this lower bound and that of convex functions is $\Omega(GD\sqrt{T})$ (Abernethy et al., 2008), we can derive the following lower bound. This corollary shows that $k$-contamination of exp-concave functions worsens regret lower bound at least $\Omega(GD\sqrt{k})$.

**Corollary 4.8.** *If $(f_1, f_2, \ldots, f_T)$ is $k$-contaminated $\alpha$-exp-concave, regret in worst case is $\Omega((d/\alpha) \log T + GD\sqrt{k})$, for all OCO algorithms.*

According to Abernethy et al. (2008), the regret lower bound in the case of $\lambda$-strongly convex functions is $\Omega((G^2/\lambda) \log T)$. Therefore, following a similar corollary is derived in the same way.

**Corollary 4.9.** *If a sequence of objective functions $(f_1, f_2, \ldots, f_T)$ is $k$-contaminated $\lambda$-strongly convex, regret in worst case is $\Omega((G^2/\lambda) \log T + GD\sqrt{k})$, for all OCO algorithms.*

## 5 Regret Upper Bounds by Universal Algorithms

In this section, we explain the regret upper bounds of some universal algorithms when the objective functions are contaminated. The algorithms we analyze in this paper are multiple eta gradient algorithm (MetaGrad) (Van Erven & Koolen, 2016), multiple sub-algorithms and learning rates (Maler) (Wang et al., 2020), and

universal strategy for online convex optimization (USC) (Zhang et al., 2022). Though there are two variations of MetaGrad; *full* MetaGrad and *diag* MetaGrad, in this paper, MetaGrad means full MetaGrad. We denote $R_T^{\boldsymbol{x}} := \sum_{t=1}^{T}(f_t(\boldsymbol{x}_t) - f_t(\boldsymbol{x}))$, $\tilde{R}_T^{\boldsymbol{x}} := \sum_{t=1}^{T} \langle \nabla f_t(\boldsymbol{x}_t), \boldsymbol{x}_t - \boldsymbol{x} \rangle$, $V_T^{\boldsymbol{x}} := \sum_{t=1}^{T}(\langle \nabla f_t(\boldsymbol{x}_t), \boldsymbol{x}_t - \boldsymbol{x} \rangle)^2$ and $W_T^{\boldsymbol{x}} := G^2 \sum_{t=1}^{T} \|\boldsymbol{x}_t - \boldsymbol{x}\|^2$.

Concerning MetaGrad and Maler, following regret bounds hold without assuming exp-concavity or strong convexity:

**Theorem 5.1.** *(Van Erven & Koolen, 2016) For MetaGrad, $R_T^{\boldsymbol{x}}$ is simultaneously bounded by $O(GD\sqrt{T \log \log T})$, and by*

$$R_T^{\boldsymbol{x}} \leq \tilde{R}_T^{\boldsymbol{x}} = O(\sqrt{V_T^{\boldsymbol{x}} d \log T} + GDd \log T),$$

*for any $\boldsymbol{x} \in \mathcal{X}$.*

**Theorem 5.2.** *(Wang et al., 2020) For Maler, $R_T^{\boldsymbol{x}}$ is simultaneously bounded by $O(GD\sqrt{T})$,*

$$R_T^{\boldsymbol{x}} \leq \tilde{R}_T^{\boldsymbol{x}} = O(\sqrt{V_T^{\boldsymbol{x}} d \log T}) \text{ and}$$
$$R_T^{\boldsymbol{x}} \leq \tilde{R}_T^{\boldsymbol{x}} = O(\sqrt{W_T^{\boldsymbol{x}} \log T}),$$

*for any $\boldsymbol{x} \in \mathcal{X}$.*

Though Theorem 5.2 is derived only for $\boldsymbol{x} \in \arg\min_{\boldsymbol{x} \in \mathcal{X}} \sum_{t=1}^{T} f_t(\boldsymbol{x})$ in the original paper by Wang et al. (2020), the proof is also valid even when $\boldsymbol{x}$ is any vector in $\mathcal{X}$, so we rewrite the statement in this form. The proof of this generalization is provided in Appendix B.5. Further explanations of universal algorithms are in Appendix A.3.

Concerning the regret bound of contaminated exp-concavity, the following theorem holds. This theorem's assumption is satisfied when using MetaGrad or Maler, and the result for them is described after the proof of this theorem.

**Theorem 5.3.** *Let $\alpha_t$ be a constant such that $f_t$ is $\alpha_t$-exp-concave (if $f_t$ is not exp-concave, then $\alpha_t$ is 0) for each t. Suppose that*

$$R_T^{\boldsymbol{x}} \leq \tilde{R}_T^{\boldsymbol{x}} = O\left(\sqrt{V_T^{\boldsymbol{x}} r_1(T)} + r_2(T)\right) \tag{1}$$

*holds for some functions $r_1$, $r_2$, and point $\boldsymbol{x} \in \mathcal{X}$. Then, it holds for any $\gamma > 0$ that*

$$R_T^{\boldsymbol{x}} = O\left(\frac{1}{\gamma} r_1(T) + GD\sqrt{k_\gamma r_1(T)} + r_2(T)\right),$$

*where $k_\gamma := \sum_{t=1}^{T} \max\{1 - \gamma_t/\gamma, 0\}$, $\gamma_t := (1/2)\min\{1/(GD), \alpha_t\}$.*

Before proving this theorem, we prepare a lemma. The proof of this lemma is given in Appendix B.6.

**Lemma 5.4.** *For all $a, b, x \geq 0$, if $x \leq \sqrt{ax} + b$, then $x \leq 3(a + b)/2$.*

*Proof of Theorem 5.3.* From Lemma 3.4, we have

$$
\begin{aligned}
R_T^{\boldsymbol{x}} &= \sum_{t=1}^{T}(f_t(\boldsymbol{x}_t) - f_t(\boldsymbol{x})) \\
&\leq \sum_{t=1}^{T}\left(\langle \nabla f_t(\boldsymbol{x}_t), \boldsymbol{x}_t - \boldsymbol{x} \rangle - \frac{\gamma_t}{2}(\langle \nabla f_t(\boldsymbol{x}_t), \boldsymbol{x} - \boldsymbol{x}_t \rangle)^2\right) \\
&= \tilde{R}_T^{\boldsymbol{x}} - \frac{\gamma}{2} V_T^{\boldsymbol{x}} + \sum_{t=1}^{T} \frac{\gamma - \gamma_t}{2}(\langle \nabla f_t(\boldsymbol{x}_t), \boldsymbol{x} - \boldsymbol{x}_t \rangle)^2 \\
&\leq \tilde{R}_T^{\boldsymbol{x}} - \frac{\gamma}{2} V_T^{\boldsymbol{x}} + \sum_{t:\gamma_t < \gamma} \frac{\gamma - \gamma_t}{2} G^2 D^2 \\
&\leq \tilde{R}_T^{\boldsymbol{x}} - \frac{\gamma}{2} V_T^{\boldsymbol{x}} + \frac{\gamma}{2} k_\gamma G^2 D^2.
\end{aligned}
$$

If $R_T^{\boldsymbol{x}} < 0$, $0$ is an upper bound, so it is sufficient to think of the case $R_T^{\boldsymbol{x}} \geq 0$. In this case, we have

$$V_T^{\boldsymbol{x}} \leq \frac{2}{\gamma} \tilde{R}_T^{\boldsymbol{x}} + k_\gamma G^2 D^2. \tag{2}$$

From equation (1), there exists a positive constant $C > 0$ such that

$$
\begin{aligned}
\tilde{R}_T^{\boldsymbol{x}} &\leq C \left( \sqrt{V_T^{\boldsymbol{x}} r_1(T)} + r_2(T) \right) \\
&\leq C \left( \sqrt{\left( \frac{2}{\gamma} \tilde{R}_T^{\boldsymbol{x}} + k_\gamma G^2 D^2 \right) r_1(T)} + r_2(T) \right) \\
&\leq \sqrt{\frac{2}{\gamma} C^2 r_1(T) \tilde{R}_T^{\boldsymbol{x}}} + CGD \sqrt{k_\gamma r_1(T)} + C r_2(T). 
\end{aligned}
\tag{3}
$$

The second inequality holds from the inequality (2), and the last inequality holds from the inequality $\sqrt{x+y} \leq \sqrt{x} + \sqrt{y}$ for $x, y \geq 0$.

From Lemma 5.4 with $a = (2/\gamma) C^2 r_1(T)$ and $b = CGD \sqrt{k_\gamma r_1(T)} + C r_2(T)$, we have

$$\tilde{R}_T^{\boldsymbol{x}} \leq \frac{3}{2} \left( \frac{2}{\gamma} C^2 r_1(T) + CGD \sqrt{k_\gamma r_1(T)} + C r_2(T) \right).$$

From this inequality and $R_T^{\boldsymbol{x}} \leq \tilde{R}_T^{\boldsymbol{x}}$, Theorem 5.3 follows. $\qquad\square$

The core of this proof is inequality (3), which can be regarded as a quadratic inequality. Solving this inequality enables us to obtain a regret upper bound for contaminated cases from a regret upper bound for non-contaminated cases.

Theorem 5.3 combined with Theorem 5.1, Theorem 5.2, and Theorem A.1 in the appendix gives upper bounds for universal algorithms; MetaGrad, Maler, and USC. The following corollary shows that, even if exp-concave objective functions are $k$-contaminated, regret can be bounded by an additional term of $O(\sqrt{kd \log T})$. This regret bound is better than ONS's in the parameter $k$.

**Corollary 5.5.** *If a sequence of objective functions* $(f_1, f_2, \ldots, f_T)$ *is* $k$-*contaminated* $\alpha$-*exp-concave, the regret bound of MetaGrad, Maler, and USC with MetaGrad or Maler as an expert algorithm is*

$$R_T = O \left( \frac{d}{\gamma} \log T + GD \sqrt{kd \log T} \right), \tag{4}$$

*where* $\gamma := (1/2) \min\{1/(GD), \alpha\}$.

We only give proof for MetaGrad and Maler here, and the proof for USC will be provided in Appendix B.7.

*Proof.* As for MetaGrad and Maler, from Theorem 5.1 and Theorem 5.2,

$$\tilde{R}_T^{\boldsymbol{x}} = O(\sqrt{V_T^{\boldsymbol{x}} d \log T} + GDd \log T)$$

holds for any $\boldsymbol{x} \in \mathcal{X}$. Therefore, by Theorem 5.3, we have

$$R_T^{\boldsymbol{x}} = O \left( \frac{d}{\gamma} \log T + GD \sqrt{k_\gamma d \log T} \right),$$

where $GD = O(1/\gamma)$ is used, which follows from $\gamma = (1/2) \min\{1/(GD), \alpha\}$. Here, $k_\gamma$ satisfies

$$k_\gamma = \sum_{t=1}^{T} \max \left\{ 1 - \frac{\gamma_t}{\gamma}, 0 \right\} = \sum_{t:\, \gamma_t < \gamma} \left( 1 - \frac{\gamma_t}{\gamma} \right) \leq k.$$

The inequality follows from the fact that if $\gamma_t < \gamma$, then $\alpha_t < \alpha$ holds. Hence, we have

$$R_T^{\boldsymbol{x}} = O\left(\frac{d}{\gamma}\log T + GD\sqrt{kd\log T}\right),$$

especially, we get the regret upper bound (4). □

As for strongly convex functions, we can get a similar result as Theorem 5.3.

**Theorem 5.6.** *Let $\lambda_t$ be a constant such that $f_t$ is $\lambda_t$-strongly convex (if $f_t$ is not strongly convex, then $\lambda_t$ is 0) for each t. Suppose that*

$$R_T^{\boldsymbol{x}} \leq \tilde{R}_T^{\boldsymbol{x}} = O\left(\sqrt{W_T^{\boldsymbol{x}} r_1(T)} + r_2(T)\right)$$

*holds for some functions $r_1$, $r_2$, and point $\boldsymbol{x} \in \mathcal{X}$. Then, it holds for any $\lambda > 0$ that*

$$R_T^{\boldsymbol{x}} = O\left(\frac{G^2}{\lambda}r_1(T) + GD\sqrt{k_\lambda r_1(T)} + r_2(T)\right),$$

*where $k_\lambda := \sum_{t=1}^T \max\{1 - \lambda_t/\lambda, 0\}$.*

This theorem can be derived in almost the same manner as the proof of Theorem 5.3, other than using the definition of strong convexity and $k_\lambda$. A more detailed proof is in Appendix B.8.

Theorem 5.6 combined with Theorem 5.1, Theorem 5.2, and Theorem A.1 in the appendix gives upper bounds for universal algorithms; MetaGrad, Maler, and USC. This corollary shows that, even if strongly convex objective functions are $k$-contaminated, regret can be bounded by an additional term of $O(\sqrt{k\log T})$ if Maler or USC with Maler as an expert algorithm is used.

**Corollary 5.7.** *If a sequence of objective functions $(f_1, f_2, \ldots, f_T)$ is $k$-contaminated $\lambda$-strongly convex, the regret bound of MetaGrad, Maler, and USC with Maler as an expert algorithm is*

$$R_T = O\left(\left(\frac{G^2}{\lambda} + GD\right)\tilde{d}\log T + GD\sqrt{k\tilde{d}\log T}\right),$$

*where $\tilde{d}$ is $d$ in the case of MetaGrad and 1 in the case of the other two algorithms.*

This corollary can be derived from Theorem 5.6 in almost the same manner as the proof of Corollary 5.5. A more detailed proof is in Appendix B.9 and Appendix B.10.

*Remark* 5.8. If $(f_1, f_2, \ldots, f_T)$ is $k_1$-contaminated $\alpha$-exp-concave and $k_2$-contaminated $\lambda$-strongly convex, then we have two regret upper bounds:

$$R_T = O\left(\frac{d}{\gamma}\log T + GD\sqrt{k_1 d\log T}\right),$$

from Corollary 5.5 and

$$R_T = O\left(\left(\frac{G^2}{\lambda} + GD\right)\tilde{d}\log T + GD\sqrt{k_2\tilde{d}\log T}\right),$$

from Corollary 5.7. Here, strongly convex functions are also exp-concave functions from Lemma 3.5. Therefore, if $\lambda/G^2 \geq \alpha$, then $k_1 \leq k_2$.

*Remark* 5.9. Note that the universal algorithms analyzed in this section do not require additional information other than the gradient, which is a valuable property in practical use. However, comparing lower bounds in Corollary 4.8 and Corollary 4.9 with upper bounds in Corollary 5.5 and Corollary 5.7 respectively, there are gaps between them. This implies that our upper bounds in this section or lower bounds in Section 4.2 might not be tight. As we will discuss in the next section, this gap can be removed if additional information on function classes are available while it is not always the case in real-world applications.

---

**Algorithm 1** Algorithm using additional information

---

**Input:** convex set $\mathcal{X} \subset \mathbb{R}^d$, $\boldsymbol{x}_1 \in \mathcal{X}$, $T$, $D$, $G$, $\alpha$, $\lambda$

1: Set $\gamma := (1/2) \min\{1/(GD), \alpha\}$, $\varepsilon := \sqrt{2}G/D$, $\boldsymbol{A}_0 := \varepsilon \boldsymbol{I}_d$.

2: **for** $t = 1$ **to** $T$ **do**

3:     Play $\boldsymbol{x}_t$ and observe cost $f_t(\boldsymbol{x}_t)$.

4:     Update:

$$\boldsymbol{A}_t = \boldsymbol{A}_{t-1} + \begin{cases} \lambda \boldsymbol{I}_d, & t \in S_1 \\ \gamma \nabla f_t(\boldsymbol{x}_t) \nabla f_t(\boldsymbol{x}_t)^\top, & t \in S_2 \\ \dfrac{G}{D\sqrt{2|[t] \cap U|}} \boldsymbol{I}_d, & t \in U, \end{cases}$$

5:     Newton step and generalized projection:

$$\boldsymbol{y}_{t+1} = \boldsymbol{x}_t - \boldsymbol{A}_t^{-1} \nabla f_t(\boldsymbol{x}_t),$$

$$\boldsymbol{x}_{t+1} = \Pi_{\mathcal{X}}^{\boldsymbol{A}_t}(\boldsymbol{y}_{t+1}) := \arg\min_{\boldsymbol{x} \in \mathcal{X}} \{\|\boldsymbol{y}_{t+1} - \boldsymbol{x}\|_{\boldsymbol{A}_t}^2\}.$$

6: **end for**

---

## 6 Regret Upper Bounds Given Additional Information

In this section, we propose a method that achieves better bounds than those of universal algorithms discussed in the previous section under the condition that the information of the class of the last objective function is revealed. The experimental performance of this method is shown in Appendix C.

We denote $S_1 := \{t \in [T] \mid f_t \text{ is } \lambda\text{-strongly convex}\}$, $S_2 := \{t \in [T]\backslash S_1 \mid f_t \text{ is } \alpha\text{-exp-concave}\}$, $U := [T]\backslash(S_1 \cup S_2)$, and $k := |U|$. The algorithm using additional information is shown in Algorithm 1 ($\boldsymbol{I}_d$ is $d$ dimensional identity matrix, and $\|\cdot\|_{\boldsymbol{A}}^2$ means $\langle \boldsymbol{A}\cdot, \cdot \rangle$). This algorithm is a generalization of OGD and ONS. Indeed, $(S_1, S_2, U) = ([T], \emptyset, \emptyset)$ gives normal OGD and $(S_1, S_2, U) = (\emptyset, [T], \emptyset)$ gives normal ONS.

Before stating the regret upper bounds of Algorithm 1, we prepare the following lemma:

**Lemma 6.1.** *Let $\{\boldsymbol{x}_t\}_t$ be the sequence generated by Algorithm 1 and $S_1$, $S_2$, $U$, and $k$ be as defined above. Then, the following inequalities hold:*

$$\sum_{t \in S_1} \|\nabla f_t(\boldsymbol{x}_t)\|_{\boldsymbol{A}_t^{-1}}^2 \leq \frac{G^2}{\lambda} \log\left(1 + \frac{\lambda D}{\sqrt{2}G}|S_1|\right), \tag{5}$$

$$\sum_{t \in S_2} \|\nabla f_t(\boldsymbol{x}_t)\|_{\boldsymbol{A}_t^{-1}}^2 \leq \frac{d}{\gamma} \log\left(1 + \frac{\lambda D}{\sqrt{2}G}|S_1| + \frac{\gamma GD}{\sqrt{2}}|S_2| + \sqrt{k}\right), \tag{6}$$

$$\sum_{t \in U} \|\nabla f_t(\boldsymbol{x}_t)\|_{\boldsymbol{A}_t^{-1}}^2 \leq \sqrt{2}GD(\sqrt{k+1} - 1). \tag{7}$$

*Proof.* Inequality (5) can be obtained as follows:

$$\sum_{t \in S_1} \|\nabla f_t(\boldsymbol{x}_t)\|_{\boldsymbol{A}_t^{-1}}^2 \leq G^2 \sum_{t \in S_1} \frac{1}{\lambda_{\min}(\boldsymbol{A}_t)} \leq G^2 \sum_{i=1}^{|S_1|} \frac{1}{\varepsilon + \lambda i} \leq G^2 \int_0^{|S_1|} \frac{\mathrm{d}s}{\varepsilon + \lambda s} = \frac{G^2}{\lambda} \log\left(1 + \frac{\lambda D}{\sqrt{2}G}|S_1|\right),$$

where $\lambda_{\min}(\boldsymbol{A}_t)$ is the minimum eigenvalue of the matrix $\boldsymbol{A}_t$, which at least increases by $\lambda$ when $t \in S_1$.

We can bound the left-hand side of the inequality (6) as follows:

$$\sum_{t \in S_2} \|\nabla f_t(\boldsymbol{x}_t)\|^2_{\boldsymbol{A}_t^{-1}} = \sum_{t \in S_2} \text{tr}\big(\boldsymbol{A}_t^{-1} \nabla f_t(\boldsymbol{x}_t)(\nabla f_t(\boldsymbol{x}_t))^\top\big)$$

$$= \frac{1}{\gamma} \sum_{t \in S_2} \text{tr}\big(\boldsymbol{A}_t^{-1}(\boldsymbol{A}_t - \boldsymbol{A}_{t-1})\big)$$

$$\leq \frac{1}{\gamma} \sum_{t \in S_2} \log \frac{|\boldsymbol{A}_t|}{|\boldsymbol{A}_{t-1}|}.$$

The first inequality is from Lemma B.5 in Appendix B.11. Since $|\boldsymbol{A}_t| \geq |\boldsymbol{A}_{t-1}|$ $(\forall t \in S_1 \cup U)$,

$$\frac{1}{\gamma} \sum_{t \in S_2} \log \frac{|\boldsymbol{A}_t|}{|\boldsymbol{A}_{t-1}|} \leq \frac{1}{\gamma} \sum_{t=1}^{T} \log \frac{|\boldsymbol{A}_t|}{|\boldsymbol{A}_{t-1}|}$$

$$= \frac{1}{\gamma} \log \frac{|\boldsymbol{A}_T|}{|\boldsymbol{A}_0|}$$

$$\leq \frac{d}{\gamma} \log \left( 1 + \frac{\lambda D}{\sqrt{2}G}|S_1| + \frac{\gamma GD}{\sqrt{2}}|S_2| + \sqrt{k} \right).$$

The last inequality is from the fact that the largest eigenvalue of $\boldsymbol{A}_T$ is at most $\sqrt{2}G/D + \lambda|S_1| + \gamma G^2|S_2| + (G/D)\sqrt{2k}$.

Inequality (7) can be obtained as follows:

$$\sum_{t \in U} \|\nabla f_t(\boldsymbol{x}_t)\|^2_{\boldsymbol{A}_t^{-1}} \leq G^2 \sum_{t \in U} \frac{1}{\lambda_{\min}(\boldsymbol{A}_t)}$$

$$\leq G^2 \sum_{t \in U} \frac{1}{\varepsilon + \sum_{i=1}^{|[t] \cap U|} \frac{G}{D\sqrt{2i}}}$$

$$\leq G^2 \sum_{t \in U} \frac{1}{\varepsilon + \sqrt{2}\frac{G}{D}(\sqrt{|[t] \cap U| + 1} - 1)}$$

$$= \frac{GD}{\sqrt{2}} \sum_{i=1}^{k} \frac{1}{\sqrt{i+1}}$$

$$\leq \sqrt{2}GD(\sqrt{k+1} - 1).$$

$\square$

Using this lemma, we can bound the regret of Algorithm 1 as follows:

**Theorem 6.2.** *Let $k$ be defined at the beginning of this section. Algorithm 1 guarantees*

$$R_T = O\left( \left( \frac{G^2}{\lambda} + \frac{d}{\gamma} \right) \log T + GD\sqrt{k} \right).$$

*Proof.* When $t \in S_1$, from the definition of strong convexity,

$$2(f_t(\boldsymbol{x}_t) - f_t(\boldsymbol{x}^*)) \leq 2 \langle \nabla f_t(\boldsymbol{x}_t), \boldsymbol{x}_t - \boldsymbol{x}^* \rangle - \lambda \|\boldsymbol{x}_t - \boldsymbol{x}^*\|^2 \tag{8}$$

holds. When $t \in S_2$, from Lemma 3.4,

$$2(f_t(\boldsymbol{x}_t) - f_t(\boldsymbol{x}^*)) \leq 2 \langle \nabla f_t(\boldsymbol{x}_t), \boldsymbol{x}_t - \boldsymbol{x}^* \rangle - \gamma(\langle \nabla f(\boldsymbol{x}_t), \boldsymbol{x}_t - \boldsymbol{x}^* \rangle)^2 \tag{9}$$

holds. When $t \in U$, since $f_t$ is convex,

$$2(f_t(\boldsymbol{x}_t) - f_t(\boldsymbol{x}^*)) \leq 2 \langle \nabla f_t(\boldsymbol{x}_t), \boldsymbol{x}_t - \boldsymbol{x}^* \rangle - \frac{G}{D\sqrt{2|[t] \cap U|}} \|\boldsymbol{x}_t - \boldsymbol{x}^*\|^2 + \frac{G}{D\sqrt{2|[t] \cap U|}} \|\boldsymbol{x}_t - \boldsymbol{x}^*\|^2 \tag{10}$$

holds. From the update rule of $\boldsymbol{A}_t$, inequalities (8), (9), and (10) can be combined into one inequality

$$2(f_t(\boldsymbol{x}_t) - f_t(\boldsymbol{x}^*)) \leq 2\langle \nabla f_t(\boldsymbol{x}_t), \boldsymbol{x}_t - \boldsymbol{x}^* \rangle - \langle (\boldsymbol{A}_t - \boldsymbol{A}_{t-1})(\boldsymbol{x}_t - \boldsymbol{x}^*), \boldsymbol{x}_t - \boldsymbol{x}^* \rangle + \frac{G1_U(t)}{D\sqrt{2|[t] \cap U|}}\|\boldsymbol{x}_t - \boldsymbol{x}^*\|^2,$$

where $1_U$ is the indicator function, i.e., $1_U(t) = 1$ if $t \in U$, and $1_U(t) = 0$ otherwise. The first and second terms in the right-hand side can be bounded as follows:

$$\begin{aligned}
&2\langle \nabla f_t(\boldsymbol{x}_t), \boldsymbol{x}_t - \boldsymbol{x}^* \rangle - \langle (\boldsymbol{A}_t - \boldsymbol{A}_{t-1})(\boldsymbol{x}_t - \boldsymbol{x}^*), \boldsymbol{x}_t - \boldsymbol{x}^* \rangle \\
&= 2\langle \boldsymbol{A}_t(\boldsymbol{y}_{t+1} - \boldsymbol{x}_t), \boldsymbol{x}^* - \boldsymbol{x}_t \rangle - \|\boldsymbol{x}_t - \boldsymbol{x}^*\|^2_{\boldsymbol{A}_t} + \|\boldsymbol{x}_t - \boldsymbol{x}^*\|^2_{\boldsymbol{A}_{t-1}} \\
&= \|\boldsymbol{y}_{t+1} - \boldsymbol{x}_t\|^2_{\boldsymbol{A}_t} - \|\boldsymbol{y}_{t+1} - \boldsymbol{x}^*\|^2_{\boldsymbol{A}_t} + \|\boldsymbol{x}_t - \boldsymbol{x}^*\|^2_{\boldsymbol{A}_{t-1}} \\
&\leq \|\boldsymbol{y}_{t+1} - \boldsymbol{x}_t\|^2_{\boldsymbol{A}_t} - \|\boldsymbol{x}_{t+1} - \boldsymbol{x}^*\|^2_{\boldsymbol{A}_t} + \|\boldsymbol{x}_t - \boldsymbol{x}^*\|^2_{\boldsymbol{A}_{t-1}}.
\end{aligned}$$

The first equality is from the algorithm, the second equality is from the law of cosines, and the last inequality is from the nonexpansiveness of projection. Therefore, we have

$$2(f_t(\boldsymbol{x}_t) - f_t(\boldsymbol{x}^*)) \leq \|\boldsymbol{y}_{t+1} - \boldsymbol{x}_t\|^2_{\boldsymbol{A}_t} - \|\boldsymbol{x}_{t+1} - \boldsymbol{x}^*\|^2_{\boldsymbol{A}_t} + \|\boldsymbol{x}_t - \boldsymbol{x}^*\|^2_{\boldsymbol{A}_{t-1}} + \frac{G1_U(t)}{D\sqrt{2|[t] \cap U|}}\|\boldsymbol{x}_t - \boldsymbol{x}^*\|^2.$$

By summing up from $t = 1$ to $T$, we can bound regret as follows:

$$\begin{aligned}
2R_T &\leq \sum_{t=1}^{T} \|\boldsymbol{y}_{t+1} - \boldsymbol{x}_t\|^2_{\boldsymbol{A}_t} + \|\boldsymbol{x}_1 - \boldsymbol{x}^*\|^2_{\boldsymbol{A}_0} + \sum_{t \in U} \frac{G}{D\sqrt{2|[t] \cap U|}}\|\boldsymbol{x}_t - \boldsymbol{x}^*\|^2 \\
&\leq \sum_{t=1}^{T} \|\nabla f_t(\boldsymbol{x}_t)\|^2_{\boldsymbol{A}_t^{-1}} + D^2\varepsilon + \frac{GD}{\sqrt{2}}\sum_{i=1}^{k}\frac{1}{\sqrt{i}} \\
&\leq \sum_{t \in S_1} \|\nabla f_t(\boldsymbol{x}_t)\|^2_{\boldsymbol{A}_t^{-1}} + \sum_{t \in S_2} \|\nabla f_t(\boldsymbol{x}_t)\|^2_{\boldsymbol{A}_t^{-1}} + \sum_{t \in U} \|\nabla f_t(\boldsymbol{x}_t)\|^2_{\boldsymbol{A}_t^{-1}} + \sqrt{2}GD(\sqrt{k} + 1).
\end{aligned}$$

From Lemma 6.1, we can get

$$\begin{aligned}
2R_T &\leq \frac{G^2}{\lambda}\log\left(1 + \frac{\lambda D}{\sqrt{2}G}|S_1|\right) + \frac{d}{\gamma}\log\left(1 + \frac{\lambda D}{\sqrt{2}G}|S_1| + \frac{\gamma GD}{\sqrt{2}}|S_2| + \sqrt{k}\right) + 2\sqrt{2}GD\sqrt{k+1} \\
&= O\left(\left(\frac{G^2}{\lambda} + \frac{d}{\gamma}\right)\log T + GD\sqrt{k}\right).
\end{aligned}$$

$\square$

The key point of Theorem 6.2 is that the second term of the regret upper bound is proportional to $\sqrt{k}$. Compared with Corollary 5.5, we can see that additional information improves the regret upper bound.

*Remark* 6.3. Algorithm 1 is written in a general form, and it is better to set $S_2 = \emptyset$ in the contaminated strongly convex case. This is because Algorithm 1 needs $O(d^3)$ computation to calculate $\boldsymbol{A}_t^{-1}$ if $S_2$ is nonempty. When $S_1 = \emptyset$ or $S_2 = \emptyset$, the regret bound in Theorem 6.2 is reduced to $O((d/\gamma)\log T + GD\sqrt{k})$ or $O((G^2/\lambda)\log T + GD\sqrt{k})$ respectively.

*Remark* 6.4. As mentioned in Remark 5.9, the algorithms analyzed in this section need information that is not always available in the real world. Therefore, the improved regret bound in Theorem 6.2 is theoretical, and regret bounds for universal algorithms explained in Section 5 are more important in real applications. However, the algorithm in this section has the notable advantage that its regret upper bounds match the lower bounds.

## 7 Conclusion

In this paper, we proposed a problem class for OCO, namely contaminated OCO, the property whose objective functions change in time steps. On this regime, we derived some upper bounds for existing and

proposed algorithms and some lower bounds of regret. While we successfully obtained optimal upper bounds with additional information of the function class of the last revealed objective function, there are still gaps of $O(\sqrt{d \log T})$ or $O(\sqrt{\log T})$ between the upper bound and the lower bound without additional information. One natural future research direction is to fill these gaps. We believe there is room for improvement in the upper bounds and the lower bounds seem tight. Indeed, lower bounds in this study interpolate well between tight bounds for general OCO and for (restricted) $\mathcal{F}$-OCO.

### Acknowledgments

Portions of this research were conducted during visits by the first author, Kamijima, to NEC Corporation.

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

## A  Existing Algorithms and Known Regret Bounds

This section introduces existing algorithms for OCO and their regret bounds.

### A.1  OGD Algorithm

In OCO, one of the most fundamental algorithms is online gradient descent (OGD), which is shown in Algorithm 2. An action $\boldsymbol{x}_t$ is updated by using the gradient of the point and projected onto the feasible region $\mathcal{X}$ in each step. If all the objective functions are convex and learning rates are set $\Theta(1/\sqrt{t})$, the regret is bounded by $O(\sqrt{T})$ (Zinkevich, 2003), and if all the objective functions are $\lambda$-strongly convex and learning rates are set $\Theta(1/t)$, the regret is bounded by $O((1/\lambda)\log T)$ (Hazan et al., 2006).

### A.2  ONS Algorithm

If all the objective functions are $\alpha$-exp-concave, ONS, shown in Algorithm 3, works well. This is an algorithm proposed by Hazan et al. (2006) as an online version of the offline Newton method. This algorithm needs parameters $\gamma, \varepsilon > 0$, and if $\gamma = (1/2)\min\{1/(GD), \alpha\}$ and $\varepsilon = 1/(\gamma^2 D^2)$, then the regret is bounded by $O((d/\alpha)\log T)$.

---

**Algorithm 2** Online Gradient Descent (Zinkevich, 2003)

---

**Input:** convex set $\mathcal{X} \subset \mathbb{R}^d$, $T$, $\boldsymbol{x}_1 \in \mathcal{X}$, parameters $\eta_t$

1: **for** $t = 1$ **to** $T$ **do**

2:     Play $\boldsymbol{x}_t$ and observe cost $f_t(\boldsymbol{x}_t)$.

3:     Gradient step and projection:
$$\boldsymbol{y}_{t+1} = \boldsymbol{x}_t - \eta_t \nabla f_t(\boldsymbol{x}_t),$$
$$\boldsymbol{x}_{t+1} = \Pi_{\mathcal{X}}(\boldsymbol{y}_{t+1}) := \arg\min_{\boldsymbol{x} \in \mathcal{X}}\{\|\boldsymbol{y}_{t+1} - \boldsymbol{x}\|^2\}.$$

4: **end for**

---

---

**Algorithm 3** Online Newton step (Hazan et al., 2006)

---

**Input:** convex set $\mathcal{X} \subset \mathbb{R}^d$, $T$, $\boldsymbol{x}_1 \in \mathcal{X}$, parameters $\gamma, \varepsilon > 0$, $\boldsymbol{A}_0 = \varepsilon \boldsymbol{I}_d$

1: **for** $t = 1$ **to** $T$ **do**
2:     Play $\boldsymbol{x}_t$ and observe cost $f_t(\boldsymbol{x}_t)$.
3:     Rank-1 update: $\boldsymbol{A}_t = \boldsymbol{A}_{t-1} + \nabla f_t(\boldsymbol{x}_t)(\nabla f_t(\boldsymbol{x}_t))^\top$.
4:     Newton step and generalized projection:

$$\boldsymbol{y}_{t+1} = \boldsymbol{x}_t - \gamma^{-1} \boldsymbol{A}_t^{-1} \nabla f_t(\boldsymbol{x}_t),$$

$$\boldsymbol{x}_{t+1} = \Pi_{\mathcal{X}}^{\boldsymbol{A}_t}(\boldsymbol{y}_{t+1}).$$

5: **end for**

---

---

**Algorithm 4** MetaGrad Master (Van Erven & Koolen, 2016)

---

**Input:** $T$, $G$, $D$, $C = 1 + 1/(1 + \lceil (1/2) \log_2 T \rceil)$

1: Set $\eta_i = 2^{-i}/(5GD)$, $\pi_1^{\eta_i} = C/((i+1)(i+2))$ for $i = 0, 1, \ldots, \lceil (1/2) \log_2 T \rceil$.
2: **for** $t = 1$ **to** $T$ **do**
3:     Get prediction $\boldsymbol{x}_t^{\eta_i}$ of slave for each $i$.
4:     Play $\boldsymbol{x}_t$:

$$\boldsymbol{x}_t = \frac{\sum_i \pi_t^{\eta_i} \eta_i \boldsymbol{x}_t^{\eta_i}}{\sum_i \pi_t^{\eta_i} \eta_i}.$$

5:     Update for each $i$:

$$\ell_t^{\eta_i}(\boldsymbol{x}_t^{\eta_i}) = -\eta_i \langle \boldsymbol{x}_t - \boldsymbol{x}_t^{\eta_i}, \nabla f_t(\boldsymbol{x}_t) \rangle + \eta_i^2 (\langle \boldsymbol{x}_t - \boldsymbol{x}_t^{\eta_i}, \nabla f_t(\boldsymbol{x}_t) \rangle)^2,$$

$$\pi_{t+1}^{\eta_i} = \frac{\pi_t^{\eta_i} \mathrm{e}^{\ell_t^{\eta_i}(\boldsymbol{x}_t^{\eta_i})}}{\sum_i \pi_t^{\eta_i} \mathrm{e}^{\ell_t^{\eta_i}(\boldsymbol{x}_t^{\eta_i})}}.$$

6: **end for**

---

### A.3 Universal Algorithms

In real-world applications, it may be unknown which function class the objective functions belong to. To cope with such cases, many universal algorithms have been developed. Most universal algorithms are constructed with two types of algorithms: a meta-algorithm and expert algorithms. Each expert algorithm is an online learning algorithm and not always universal. In each time step, expert algorithms update $\boldsymbol{x}_t^i$, and a meta-algorithm integrates these outputs in some way, such as a convex combination. In the following, we explain three universal algorithms: multiple eta gradient algorithm (MetaGrad) (Van Erven & Koolen, 2016), multiple sub-algorithms and learning rates (Maler) (Wang et al., 2020), and universal strategy for online convex optimization (USC) (Zhang et al., 2022).

First, MetaGrad is an algorithm with multiple experts, each with a different parameter $\eta$ as shown in Algorithm 4 and 5. In contrast to nonuniversal algorithms that need to set parameters beforehand depending on the property of objective functions, MetaGrad sets multiple $\eta$ so that users do not need prior knowledge. It is known that MetaGrad achieves $O(\sqrt{T \log \log T})$, $O((d/\lambda) \log T)$ and $O((d/\alpha) \log T)$ regret bounds for convex, $\lambda$-strongly convex and $\alpha$-exp-concave objective functions respectively.

Second, Maler is an algorithm with three types of expert algorithms: a convex expert algorithm, strongly convex expert algorithms, and exp-concave expert algorithms, as shown in Algorithm 6 to 9. They are similar to OGD with $\Theta(1/\sqrt{t})$ stepsize, OGD with $\Theta(1/t)$ stepsize, and ONS, respectively. Expert algorithms contain multiple strongly convex expert algorithms and multiple exp-concave expert algorithms with multiple parameters $\eta$ like MetaGrad. It is known that Maler achieves $O(\sqrt{T})$, $O((1/\lambda) \log T)$ and $O((d/\alpha) \log T)$ regret bounds for convex, $\lambda$-strongly convex and $\alpha$-exp-concave objective functions respectively.

---

**Algorithm 5** MetaGrad Slave (Van Erven & Koolen, 2016)

---

**Input:** convex set $\mathcal{X} \subset \mathbb{R}^d$, $T$, $\eta$, $D$

1: Set $\boldsymbol{x}_1^\eta = \boldsymbol{0}$, $\boldsymbol{\Sigma}_1^\eta = D^2 \boldsymbol{I}_d$
2: **for** $t = 1$ **to** $T$ **do**
3:      Issue $\boldsymbol{x}_t^\eta$ to master.
4:      Update:

$$\boldsymbol{\Sigma}_{t+1}^\eta = \left( \frac{1}{D^2} \boldsymbol{I}_d + 2\eta^2 \sum_{s=1}^t \nabla f_t(\boldsymbol{x}_t)(\nabla f_t(\boldsymbol{x}_t))^\top \right)^{-1},$$

$$\tilde{\boldsymbol{x}}_{t+1}^\eta = \boldsymbol{x}_t^\eta - \eta \boldsymbol{\Sigma}_{t+1}^\eta (1 + 2\eta^2 \langle \nabla f_t(\boldsymbol{x}_t), \boldsymbol{x}_t^\eta - \boldsymbol{x}_t \rangle) \nabla f_t(\boldsymbol{x}_t),$$

$$\boldsymbol{x}_{t+1}^\eta = \Pi_{\mathcal{X}}^{(\boldsymbol{\Sigma}_{t+1}^\eta)^{-1}} (\tilde{\boldsymbol{x}}_{t+1}^\eta).$$

5: **end for**

---

---

**Algorithm 6** Maler Meta (Wang et al., 2020)

---

**Input:** $T$, $G$, $D$, $C = 1 + 1/(1 + \lceil (1/2) \log_2 T \rceil)$

1: Set $\eta^c = 1/(2GD\sqrt{T})$, $\eta_i = 2^{-i}/(5GD)$ for $i = 0, 1, \ldots, \lceil (1/2) \log_2 T \rceil$.
2: Set $\pi^c = 1/3$, $\pi_1^{\eta_i, \ell} = \pi_1^{\eta_i, s} = C/(3(i+1)(i+2))$ for $i = 0, 1, \ldots, \lceil (1/2) \log_2 T \rceil$.
3: **for** $t = 1$ **to** $T$ **do**
4:      Get predictions $\boldsymbol{x}_t^c$ from Algorithm 7 and $\boldsymbol{x}_t^{\eta_i, \ell}, \boldsymbol{x}_t^{\eta_i, s}$ from Algorithms 8 and 9 for all $i$.
5:      Play

$$\boldsymbol{x}_t = \frac{\pi_t^c \eta^c \boldsymbol{x}_t^c + \sum_i (\pi_t^{\eta_i, s} \eta_i \boldsymbol{x}_t^{\eta_i, s} + \pi_t^{\eta_i, \ell} \eta_i \boldsymbol{x}_t^{\eta_i, \ell})}{\pi_t^c \eta^c + \sum_i (\pi_t^{\eta_i, s} \eta_i + \pi_t^{\eta_i, \ell} \eta_i)}.$$

6:      Observe gradient $\nabla f_t(\boldsymbol{x}_t)$ and send it to all experts.
7:      Update weights:

$$\pi_{t+1}^c = \frac{\pi_t^c \mathrm{e}^{-c_t(\boldsymbol{x}_t^c)}}{\Phi_t},$$

$$\pi_{t+1}^{\eta_i, s} = \frac{\pi_t^{\eta_i, s} \mathrm{e}^{-s_t^{\eta_i}(\boldsymbol{x}_t^{\eta_i, s})}}{\Phi_t} \text{ for each } i,$$

$$\pi_{t+1}^{\eta_i, \ell} = \frac{\pi_t^{\eta_i, \ell} \mathrm{e}^{-\ell_t^{\eta_i}(\boldsymbol{x}_t^{\eta_i, \ell})}}{\Phi_t} \text{ for each } i,$$

     where

$$\Phi_t = \sum_i (\pi_1^{\eta_i, s} \mathrm{e}^{-\sum_{\tau=1}^t s_\tau^{\eta_i}(\boldsymbol{x}_\tau^{\eta_i, s})} + \pi_1^{\eta_i, \ell} \mathrm{e}^{-\sum_{\tau=1}^t \ell_\tau^{\eta_i}(\boldsymbol{x}_\tau^{\eta_i, \ell})}) + \pi_1^c \mathrm{e}^{-\sum_{\tau=1}^t c_\tau(\boldsymbol{x}_\tau^c)},$$

$$c_t(\boldsymbol{x}) = -\eta^c \langle \nabla f_t(\boldsymbol{x}_t), \boldsymbol{x}_t - \boldsymbol{x} \rangle + (\eta^c GD)^2,$$

$$s_t^\eta(\boldsymbol{x}) = -\eta \langle \nabla f_t(\boldsymbol{x}_t), \boldsymbol{x}_t - \boldsymbol{x} \rangle + \eta^2 G^2 \|\boldsymbol{x}_t - \boldsymbol{x}\|^2,$$

$$\ell_t^\eta(\boldsymbol{x}) = -\eta \langle \nabla f_t(\boldsymbol{x}_t), \boldsymbol{x}_t - \boldsymbol{x} \rangle + \eta^2 (\langle \nabla f_t(\boldsymbol{x}_t), \boldsymbol{x}_t - \boldsymbol{x} \rangle)^2.$$

8: **end for**

---

Finally, USC is an algorithm with many expert algorithms, as shown in Algorithm 10. In contrast to Maler, which contains OGD and ONS as expert algorithms, USC contains more expert algorithms. To integrate many experts, USC utilizes Adapt-ML-Prod (Gaillard et al., 2014) as a meta-algorithm, which realizes universal regret bound. Concerning the regret bound of USC, there is a theorem as follows.

---

**Algorithm 7** Maler Convex Expert (Wang et al., 2020)

---

**Input:** convex set $\mathcal{X} \subset \mathbb{R}^d$, $T$, $G$, $D$, $\eta^c$
1: Set $\boldsymbol{x}_t^c = \boldsymbol{0}$.
2: **for** $t = 1$ **to** $T$ **do**
3:   Send $\boldsymbol{x}_t^c$ to Algorithm 6.
4:   Receive gradient $\nabla f_t(\boldsymbol{x}_t)$ from Algorithm 6.
5:   Update:
$$\boldsymbol{x}_{t+1}^c = \Pi_{\mathcal{X}} \left( \boldsymbol{x}_t^c - \frac{D}{\eta^c G \sqrt{t}} \nabla c_t(\boldsymbol{x}_t^c) \right).$$

6: **end for**

---

**Algorithm 8** Maler Exp-concave Expert (Wang et al., 2020)

---

**Input:** convex set $\mathcal{X} \subset \mathbb{R}^d$, $T$, $D$, $\eta$
1: Set $\boldsymbol{x}_t^{\eta,\ell} = \boldsymbol{0}$, $\beta = 1/2$, $\boldsymbol{\Sigma}_1 = (1/(\beta^2 D^2)) \boldsymbol{I}_d$.
2: **for** $t = 1$ **to** $T$ **do**
3:   Send $\boldsymbol{x}_t^{\eta,\ell}$ to Algorithm 6.
4:   Receive gradient $\nabla f_t(\boldsymbol{x}_t)$ from Algorithm 6.
5:   Update:
$$\boldsymbol{\Sigma}_{t+1} = \boldsymbol{\Sigma}_t + \nabla \ell_t^{\eta}(\boldsymbol{x}_t^{\eta,\ell})(\nabla \ell_t^{\eta}(\boldsymbol{x}_t^{\eta,\ell}))^{\top},$$
$$\boldsymbol{x}_{t+1}^{\eta,\ell} = \Pi_{\mathcal{X}}^{\boldsymbol{\Sigma}_{t+1}} \left( \boldsymbol{x}_t^{\eta,\ell} - \frac{1}{\beta} \boldsymbol{\Sigma}_{t+1}^{-1} \nabla \ell_t^{\eta}(\boldsymbol{x}_t^{\eta,\ell}) \right).$$

6: **end for**

---

**Theorem A.1.** *(Zhang et al., 2022) Let $\mathcal{E}$ be a set of expert algorithms and $\boldsymbol{x}_t^i$ be an output of $i$th algorithm in $t$ time step. Then,*

$$\sum_{t=1}^{T} (f_t(\boldsymbol{x}_t) - f_t(\boldsymbol{x}_t^i)) \leq \sum_{t=1}^{T} \left\langle \nabla f_t(\boldsymbol{x}_t), \boldsymbol{x}_t - \boldsymbol{x}_t^i \right\rangle \leq 4\Gamma GD + \frac{\Gamma}{\sqrt{\log |\mathcal{E}|}} \sqrt{4G^2 D^2 + \sum_{t=1}^{T} (\langle \nabla f_t(\boldsymbol{x}_t), \boldsymbol{x}_t - \boldsymbol{x}_t^i \rangle)^2},$$

*where $\Gamma = O(\log \log T)$.*

In USC, expert algorithms are chosen so that $|\mathcal{E}| = O(\log T)$ holds. This theorem holds without assuming exp-concavity or strong convexity. In addition, it is known that USC achieves $O(\sqrt{L_T \log \log T})$, $O((1/\lambda) \cdot (\min\{\log L_T, \log V_T\} + \log \log T))$ and $O((1/\alpha)(d \min\{\log L_T, \log V_T\} + \log \log T))$ regret bounds for convex, $\lambda$-strongly convex and $\alpha$-exp-concave objective functions respectively, where $L_T := \min_{\boldsymbol{x} \in \mathcal{X}} \sum_{t=1}^{T} f_t(\boldsymbol{x}) = O(T)$, $V_T := \sum_{t=1}^{T} \max_{\boldsymbol{x} \in \mathcal{X}} \|\nabla f_t(\boldsymbol{x}) - \nabla f_{t-1}(\boldsymbol{x})\|_2^2 = O(T)$.

---

**Algorithm 9** Maler Strongly Convex Expert (Wang et al., 2020)

---

**Input:** convex set $\mathcal{X} \subset \mathbb{R}^d$, $T$, $G$, $\eta$
1: Set $\boldsymbol{x}_t^{\eta,s} = \boldsymbol{0}$.
2: **for** $t = 1$ **to** $T$ **do**
3:   Send $\boldsymbol{x}_t^{\eta,s}$ to Algorithm 6.
4:   Receive gradient $\nabla f_t(\boldsymbol{x}_t)$ from Algorithm 6.
5:   Update:
$$\boldsymbol{x}_{t+1}^{\eta,s} = \Pi_{\mathcal{X}} \left( \boldsymbol{x}_t^{\eta,s} - \frac{1}{2\eta^2 G^2 t} \nabla s_t^{\eta}(\boldsymbol{x}_t^{\eta,s}) \right).$$

6: **end for**

---

---

**Algorithm 10** Universal Strategy for Online Convex Optimization (USC) (Zhang et al., 2022)

---

**Input:** $\mathcal{A}_{str}$, $\mathcal{A}_{exp}$, and $\mathcal{A}_{con}$, which are sets of algorithms designed for strongly convex functions, exp-concave functions and general convex functions respectively; $\mathcal{P}_{str}$ and $\mathcal{P}_{exp}$, which are sets of parameters of strong convexity and exp-concavity respectively.

1: Initialize $\mathcal{E} = \emptyset$.
2: **for** each algorithm $A \in \mathcal{A}_{str}$ **do**
3:     **for** each $\lambda \in \mathcal{P}_{str}$ **do**
4:         Create an expert $E(A, \lambda)$.
5:         Update $\mathcal{E} = \mathcal{E} \cup E(A, \lambda)$.
6:     **end for**
7: **end for**
8: **for** each algorithm $A \in \mathcal{A}_{exp}$ **do**
9:     **for** each $\alpha \in \mathcal{P}_{exp}$ **do**
10:        Create an expert $E(A, \alpha)$.
11:        Update $\mathcal{E} = \mathcal{E} \cup E(A, \alpha)$.
12:     **end for**
13: **end for**
14: **for** each algorithm $A \in \mathcal{A}_{con}$ **do**
15:     Create an expert $E(A)$.
16:     Update $\mathcal{E} = \mathcal{E} \cup E(A)$.
17: **end for**
18: **for** $t = 1$ **to** $T$ **do**
19:     Calculate the weight $p_t^i$ of each expert $E^i$ by

$$p_t^i = \frac{\eta_{t-1}^i w_{t-1}^i}{\sum_{j=1}^{|\mathcal{E}|} \eta_{t-1}^j w_{t-1}^j}.$$

20:     Receive $\boldsymbol{x}_t^i$ from each expert $E^i \in \mathcal{E}$.
21:     Output the weighted average $\boldsymbol{x}_t = \sum_{i=1}^{|\mathcal{E}|} p_t^i \boldsymbol{x}_t^i$.
22:     Observe the loss function $f_t(\cdot)$.
23:     Send the function $f_t(\cdot)$ to each expert $E^i \in \mathcal{E}$.
24: **end for**

---

# B MISSING PROOFS

In this section, we explain missing proofs.

## B.1 Proof of the Exp-Concavity of the Function in Example 3.8

In this subsection, we present the proof of the exp-concavity of the function $f_t$ in Example 3.8 in the case $a_{t,i} = a_t$, $b_{t,i} = b_t$. Before the proof, we introduce the following lemma.

**Lemma B.1.** *(Hazan, 2016) A twice-differentiable function $f \colon \mathbb{R}^d \to \mathbb{R}$ is $\alpha$-exp-concave at $\boldsymbol{x}$ if and only if*

$$\nabla^2 f(\boldsymbol{x}) \succeq \alpha \nabla f(\boldsymbol{x}) \nabla f(\boldsymbol{x})^\top.$$

Using this lemma, we can check the exp-concavity of the function $f_t$.

*Proof.* By differentiating $f_t$, we have

$$\nabla f_t(\boldsymbol{x}) = -\frac{b_t \boldsymbol{a}_t}{1 + \exp(b_t \langle \boldsymbol{a}_t, \boldsymbol{x} \rangle)}, \quad \nabla^2 f_t(\boldsymbol{x}) = \frac{b_t^2 \boldsymbol{a}_t \boldsymbol{a}_t^\top \exp(b_t \langle \boldsymbol{a}_t, \boldsymbol{x} \rangle)}{(1 + \exp(b_t \langle \boldsymbol{a}_t, \boldsymbol{x} \rangle))^2}.$$

For all $\boldsymbol{v} \in \mathbb{R}^d$,

$$\boldsymbol{v}^{\top}(\nabla^2 f_t(\boldsymbol{x}) - \alpha \nabla f_t(\boldsymbol{x}) \nabla f_t(\boldsymbol{x})^{\top})\boldsymbol{v} = b_t^2 \langle \boldsymbol{a}_t, \boldsymbol{v} \rangle^2 \frac{\exp(b_t \langle \boldsymbol{a}_t, \boldsymbol{x} \rangle) - \alpha}{(1 + \exp(b_t \langle \boldsymbol{a}_t, \boldsymbol{x} \rangle))^2}$$

holds, and combined with Lemma B.1, $f_t$ is $\exp(-\|\boldsymbol{a}_t\|)$-exp-concave. □

## B.2 Proof of Proposition 4.1

This subsection presents the proof of Proposition 4.1.

*Proof.* Let $\boldsymbol{x}^* \in \arg\min_{\boldsymbol{x} \in \mathcal{X}} \sum_{t=1}^{T} f_t(\boldsymbol{x})$. We can bound regret as follows:

$$\begin{aligned}
R_T &= \sum_{t=1}^{T} (f_t(\boldsymbol{x}_t) - f_t(\boldsymbol{x}^*)) \\
&\leq \sum_{t=1}^{T} \left( \langle \nabla f_t(\boldsymbol{x}_t), \boldsymbol{x}_t - \boldsymbol{x}^* \rangle - \frac{\gamma_t}{2} (\langle \nabla f_t(\boldsymbol{x}_t), \boldsymbol{x}_t - \boldsymbol{x}^* \rangle)^2 \right) \\
&= \sum_{t=1}^{T} \left( \langle \nabla f_t(\boldsymbol{x}_t), \boldsymbol{x}_t - \boldsymbol{x}^* \rangle - \frac{\gamma}{2} (\langle \nabla f_t(\boldsymbol{x}_t), \boldsymbol{x}_t - \boldsymbol{x}^* \rangle)^2 \right) + \sum_{t=1}^{T} \frac{\gamma - \gamma_t}{2} (\langle \nabla f_t(\boldsymbol{x}_t), \boldsymbol{x}_t - \boldsymbol{x}^* \rangle)^2 \\
&\leq \sum_{t=1}^{T} \left( \langle \nabla f_t(\boldsymbol{x}_t), \boldsymbol{x}_t - \boldsymbol{x}^* \rangle - \frac{\gamma}{2} (\langle \nabla f_t(\boldsymbol{x}_t), \boldsymbol{x}_t - \boldsymbol{x}^* \rangle)^2 \right) + \sum_{t:\gamma_t < \gamma} \frac{\gamma - \gamma_t}{2} G^2 D^2 \\
&\leq \frac{2d}{\gamma} \log T + \frac{1}{4} k G D,
\end{aligned}$$

where $\gamma_t$ is defined in the same way as defined in Theorem 5.3. The first inequality is from Lemma 3.4. In the last inequality, the first term is bounded by $(2d/\gamma) \log T$ because of the proof of ONS's regret bound by Hazan (2016). The second term is bounded by $(1/4)kGD$ from $\gamma \leq 1/(2GD)$ by definition of $\gamma$. □

## B.3 Proof of Theorem 4.2

This subsection presents the proof of Theorem 4.2.

*Proof.* Let $I_1$ and $I_2$ be instances used to prove lower bound $R_T = \Omega(g_1(T))$ for function class $\mathcal{F}$ and $R_k = \Omega(g_2(k))$ for convex objective functions, respectively, and $f_{i,t}$ ($i = 1, 2$) be objective functions of $I_i$ at time step $t$, and $\mathcal{X}_i$ be sets which decision variables of $I_i$ belong to. Here, take a set $\mathcal{X}$ so that there exist surjections $\phi_i \colon \mathcal{X} \to \mathcal{X}_i$. For this $\mathcal{X}$, let $\tilde{I}_1$ be an instance whose objective function at time step $t$ is $f_{1,t} \circ \phi_1$ and $\tilde{I}_2$ be an instance whose objective function at time step $t$ is $f_{2,t} \circ \phi_2$ if $t \leq k$, and some function in $\mathcal{F}$ whose minimizer is the same as the minimizer of $\sum_{t=1}^{k} f_{2,t}$ otherwise. For these instances, consider the case that instances $\tilde{I}_1$ and $\tilde{I}_2$ realize with probability $1/2$. In this case, the expectation of regret satisfies

$$\begin{aligned}
\mathbb{E}[R_T] &= \frac{1}{2} \Omega(g_1(T)) + \frac{1}{2} \Omega(g_2(k)) \\
&= \Omega(g_1(T) + g_2(k)),
\end{aligned}$$

for all OCO algorithms. Therefore, Theorem 4.2 follows. □

## B.4 Proof of Proposition 4.4

This subsection presents the proof of Proposition 4.4.

*Proof.* Consider the instance as follows:

$$f_t(x) = v_t x, \ x \in \mathcal{X} = [-D/2, D/2], \ x_1 = -D/2,$$

where

$$
v_t = \begin{cases} (-1)^t G & t < t_1, \\ G & t \geq t_1, \ x_{t_1} \geq 0, \\ -G & t \geq t_1, \ x_{t_1} < 0, \end{cases}
$$

and $t_1$ is a minimum natural number which satisfies $t_1 \geq (1 + \gamma G^2 D/2)^{-1} T$. Then,

$$
\min_{x \in \mathcal{X}} \sum_{t=1}^{T} f_t(x) \leq (-T + t_1) \frac{GD}{2} \leq \left( -\frac{\gamma G^2 D/2}{1 + \gamma G^2 D/2} T + 1 \right) \frac{GD}{2}. \tag{11}
$$

The second inequality is from $t_1 \leq (1 + \gamma G^2 D/2)^{-1} T + 1$. If $x_{t_1} \geq 0$,

$$
\sum_{t=1}^{T} f_t(x_t) = G \sum_{t=1}^{t_1-1} (-1)^t x_t + G \sum_{t=t_1}^{T} x_t. \tag{12}
$$

For the first term, since

$$
A_t = \varepsilon + G^2 t
$$

for all $t \in [T]$, and if $t < t_1$, we have

$$
y_{t+1} = x_t - \gamma^{-1} (\varepsilon + G^2 t)^{-1} (-1)^t G.
$$

Now, $x_{t+1}$ is defined as

$$
x_{t+1} = \begin{cases} \dfrac{D}{2} & y_{t+1} > \dfrac{D}{2}, \\ y_{t+1} & -\dfrac{D}{2} \leq y_{t+1} \leq \dfrac{D}{2}, \\ -\dfrac{D}{2} & y_{t+1} < -\dfrac{D}{2}, \end{cases}
$$

and therefore, we get

$$
x_t = \begin{cases} (-1)^t \dfrac{D}{2} & t \leq t_2, \\ (-1)^{t_2} \dfrac{D}{2} - \displaystyle\sum_{s=t_2}^{t-1} \dfrac{(-1)^s G}{\gamma(\varepsilon + G^2 s)} & t > t_2, \end{cases}
$$

where $t_2$ is a minimum time step $t$ which satisfies $G\gamma^{-1}(\varepsilon + G^2 t)^{-1} < D$, i.e., $t > 1/(\gamma G D) - \varepsilon/G^2$. From this, for sufficiently large $T$ so that $t_2 \leq t_1 - 2$, we have

$$
\begin{aligned}
G \sum_{t=1}^{t_1-1} (-1)^t x_t &= G \sum_{t=1}^{t_2} (-1)^t (-1)^t \frac{D}{2} + \sum_{t=t_2+1}^{t_1-1} (-1)^t \left( (-1)^{t_2} \frac{D}{2} - \sum_{s=t_2}^{t-1} \frac{(-1)^s G}{\gamma(\varepsilon + G^2 s)} \right) \\
&\geq \frac{t_2 - 1}{2} GD + \frac{G^2}{\gamma} \sum_{t=t_2+1}^{t_1-1} \sum_{s=t_2}^{t-1} \frac{(-1)^{s+t+1}}{\varepsilon + G^2 s} \\
&= \frac{t_2 - 1}{2} GD + \frac{G^2}{\gamma} \sum_{s=t_2}^{t_1-2} \sum_{t=s+1}^{t_1-1} \frac{(-1)^{s+t+1}}{\varepsilon + G^2 s} \\
&\geq \frac{t_2 - 1}{2} GD - \frac{G^2}{\gamma} \sum_{s=t_2}^{t_1-2} \frac{1}{\varepsilon + G^2 s} \\
&\geq \frac{t_2 - 1}{2} GD - \frac{G^2}{\gamma} \int_{t_2-1}^{t_1-2} \frac{\mathrm{d}s}{\varepsilon + G^2 s} \\
&= \frac{t_2 - 1}{2} GD - \frac{1}{\gamma} \log \frac{\varepsilon + (t_1 - 2)G^2}{\varepsilon + (t_2 - 1)G^2} \\
&\geq -\frac{1}{\gamma} \log \left( 1 + \frac{G^2}{\varepsilon} T \right). \tag{13}
\end{aligned}
$$

Next, for the second term of equation (12), if $x_{t_1} \geq 0$ and $t \geq t_1$, we then have

$$y_{t+1} = x_t - \gamma^{-1}(\varepsilon + G^2 t)^{-1} \geq x_t - \gamma^{-1}G^{-2}t_1^{-1}$$

and since $x_{t+1} \geq y_{t+1}$ holds from $y_{t+1} \leq x_t \leq D/2$, we have

$$\begin{aligned}
y_{t+1} &\geq x_{t_1} - (t - t_1)\gamma^{-1}G^{-2}t_1^{-1} \\
&\geq -(T - t_1)\gamma^{-1}G^{-2}t_1^{-1} \\
&= -\left(\frac{T}{t_1} - 1\right)G^{-2}\gamma^{-1} \\
&\geq -\left(\frac{T}{(1 + \gamma G^2 D/2)^{-1}T} - 1\right)G^{-2}\gamma^{-1} \\
&= -\frac{D}{2}.
\end{aligned}$$

Therefore, from

$$x_t = x_{t_1} - \sum_{s=t_1}^{t-1}\gamma^{-1}(\varepsilon + G^2 s)^{-1},$$

we have

$$\begin{aligned}
G\sum_{t=t_1}^{T}x_t &= G\sum_{t=t_1}^{T}\left(x_{t_1} - \sum_{s=t_1}^{t-1}\gamma^{-1}(\varepsilon + G^2 s)^{-1}\right) \\
&\geq -\frac{1}{\gamma G}\sum_{t=t_1}^{T}\sum_{s=t_1}^{t-1}s^{-1} \\
&= -\frac{1}{\gamma G}\sum_{s=t_1}^{T-1}\sum_{t=s+1}^{T}s^{-1} \\
&= -\frac{1}{\gamma G}\sum_{s=t_1}^{T-1}\left(-1 + \frac{T}{s}\right) \\
&\geq \frac{T - t_1 - 1}{\gamma G} - \frac{T}{\gamma G}\log\frac{T}{t_1} \\
&\geq \frac{T - (1 + \gamma G^2 D/2)^{-1}T - 2}{\gamma G} - \frac{T}{\gamma G}\log\frac{T}{(1 + \gamma G^2 D/2)^{-1}T} \\
&= \frac{T}{G}\left(\frac{G^2 D/2}{1 + \gamma G^2 D/2} - \gamma^{-1}\log\left(1 + \gamma G^2 D/2\right)\right) - \frac{2}{\gamma G}.
\end{aligned} \tag{14}$$

We can derive the same bound similarly in the case of $x_{t_1} < 0$.

From inequality (11), equality (12), inequality (13) and inequality (14), we complete the proof:

$$\begin{aligned}
R_T &\geq -\frac{1}{\gamma}\log\left(1 + \frac{G^2}{\varepsilon}T\right) + \frac{T}{G}\left(\frac{G^2 D/2}{1 + \gamma G^2 D/2} - \gamma^{-1}\log\left(1 + \gamma G^2 D/2\right)\right) - \frac{2}{\gamma G} \\
&\quad - \left(-\frac{\gamma G^2 D/2}{1 + \gamma G^2 D/2}T + 1\right)\frac{GD}{2} \\
&\geq T\left(\frac{G^2 D}{2} - \gamma^{-1}\left(\gamma\frac{G^2 D}{2} - \frac{(\gamma G^2 D/2)^2}{2(1 + \gamma G^2 D/2)^2}\right)\right) - \frac{1}{\gamma}\log\left(1 + \frac{G^2}{\varepsilon}T\right) - \frac{2}{\gamma G} - \frac{GD}{2} \\
&= \frac{\gamma G^2 D/2}{2(1 + \gamma G^2 D/2)^2}T - \frac{1}{\gamma}\log\left(1 + \frac{G^2}{\varepsilon}T\right) - \frac{2}{\gamma G} - \frac{G^2 D}{2} \\
&= \Omega(T).
\end{aligned}$$

The second inequality follows from the inequality $\log\left(1 + \gamma G^2 D/2\right) \leq \gamma G^2 D/2 - \frac{(\gamma G^2 D/2)^2}{2(1 + \gamma G^2 D/2)^2}$ for any $\gamma > 0$ by Taylor's theorem. $\qquad\square$

### B.5 Proof of Theorem 5.2

This subsection proves that Theorem 5.2 holds for any $\boldsymbol{x} \in \mathcal{X}$. Before stating the proof of Theorem 5.2, we introduce two following lemmas. Here, $c_t$, $\boldsymbol{x}_t^c$, $\ell_t^\eta$, $\boldsymbol{x}_t^{\eta,\ell}$, $s_t^\eta$, and $\boldsymbol{x}_t^{\eta,s}$ are introduced in Algorithm 6.

**Lemma B.2.** *(Wang et al., 2020) For every grid point $\eta$, we have*

$$\sum_{t=1}^{T}(c_t(\boldsymbol{x}_t) - c_t(\boldsymbol{x}_t^c)) \le \log 3 + \frac{1}{4}, \tag{15}$$

$$\sum_{t=1}^{T}(\ell_t^\eta(\boldsymbol{x}_t) - \ell_t^\eta(\boldsymbol{x}_t^{\eta,\ell})) \le 2\log\left(\sqrt{3}\left(\frac{1}{2}\log_2 T + 3\right)\right), \tag{16}$$

*and*

$$\sum_{t=1}^{T}(s_t^\eta(\boldsymbol{x}_t) - s_t^\eta(\boldsymbol{x}_t^{\eta,s})) \le 2\log\left(\sqrt{3}\left(\frac{1}{2}\log_2 T + 3\right)\right). \tag{17}$$

*Remark* B.3. According to Wang et al. (2020),

$$\sum_{t=1}^{T}(c_t(\boldsymbol{x}_t) - c_t(\boldsymbol{x}_t^c)) \le \log 3$$

holds instead of inequality (15). However, the above inequality needs to be corrected. In the last part of the proof of this lemma in their paper, they derived

$$0 \le \sum_{t=1}^{T} c_t(\boldsymbol{x}_t^c) + \log\frac{1}{\pi^c} = \sum_{t=1}^{T} c_t(\boldsymbol{x}_t^c) + \log 3.$$

From the definition of $c_t$ and $\eta^c = 1/(2GD\sqrt{T})$, we have

$$\sum_{t=1}^{T} c_t(\boldsymbol{x}_t) = \sum_{t=1}^{T}(\eta^c GD)^2 = \sum_{t=1}^{T}\frac{1}{4T} = \frac{1}{4},$$

though they treated this term as 0. Combining these relationships, we get inequality (15). This mistake seems to be a mere typo since the regret bound in their paper coincides with the result derived from the inequality (15).

**Lemma B.4.** *(Wang et al., 2020) For every grid point $\eta$ and any $\boldsymbol{x} \in \mathcal{X}$, we have*

$$\sum_{t=1}^{T}(c_t(\boldsymbol{x}_t^c) - c_t^\eta(\boldsymbol{x})) \le \frac{3}{4}, \tag{18}$$

$$\sum_{t=1}^{T}(\ell_t^\eta(\boldsymbol{x}_t^{\eta,\ell}) - \ell_t^\eta(\boldsymbol{x})) \le 10 d\log T, \tag{19}$$

*and*

$$\sum_{t=1}^{T}(s_t^\eta(\boldsymbol{x}_t^{\eta,s}) - s_t^\eta(\boldsymbol{x})) \le 1 + \log T. \tag{20}$$

*Proof of Theorem 5.2.* We can get $O(GD\sqrt{T})$ bound as follows:

$$R_T^{\boldsymbol{x}} \le \tilde{R}_T^{\boldsymbol{x}} = \frac{1}{\eta^c}\sum_{t=1}^{T}((\eta^c GD)^2 - c_t(\boldsymbol{x}))$$

$$= \frac{1}{\eta^c}\left(\sum_{t=1}^{T}(c_t(\boldsymbol{x}_t) - c_t(\boldsymbol{x}_t^c)) + \sum_{t=1}^{T}(c_t(\boldsymbol{x}_t^c) - c_t(\boldsymbol{x}))\right)$$

$$\le 2(1 + \log 3)GD\sqrt{T},$$

where the last inequality follows from inequalities (15) and (18).

We can get $O(\sqrt{W_T^{\boldsymbol{x}} \log T})$ bound as follows:

$$
\begin{aligned}
R_T^{\boldsymbol{x}} \leq \tilde{R}_T^{\boldsymbol{x}} &= \frac{1}{\eta} \sum_{t=1}^{T} ((\eta G)^2 \|\boldsymbol{x}_t - \boldsymbol{x}\|^2 - s_t^{\eta}(\boldsymbol{x})) \\
&= \eta \sum_{t=1}^{T} G^2 \|\boldsymbol{x}_t - \boldsymbol{x}\|^2 + \frac{1}{\eta} \left( \sum_{t=1}^{T} (s_t(\boldsymbol{x}_t) - s_t(\boldsymbol{x}_t^s)) + \sum_{t=1}^{T} (s_t(\boldsymbol{x}_t^s) - s_t(\boldsymbol{x})) \right) \\
&\leq \eta W_T^{\boldsymbol{x}} + \frac{A}{\eta},
\end{aligned}
$$

where $A := 2\log\left(\sqrt{3}((1/2)\log_2 T + 3)\right) + 1 + \log T \geq 1$ and the last inequality follows from inequalities (17) and (20). Let $\hat{\eta} := \sqrt{A/W_T^{\boldsymbol{x}}} \geq 1/(5GD\sqrt{T})$. If $\hat{\eta} \leq 1/(5GD)$, there exists a grid point $\eta_i \in [\hat{\eta}/2, \hat{\eta}]$ and we get

$$
\tilde{R}_T^{\boldsymbol{x}} \leq \eta_i W_T^{\boldsymbol{x}} + \frac{A}{\eta_i} \leq \hat{\eta} W_T^{\boldsymbol{x}} + \frac{2A}{\hat{\eta}} = \sqrt{W_T^{\boldsymbol{x}} A}.
$$

Otherwise, since $W_T^{\boldsymbol{x}} \leq 25G^2 D^2 A$, by taking $\eta = \eta_1 = 1/(5GD)$, we get

$$
\tilde{R}_T^{\boldsymbol{x}} \leq \eta_1 W_T^{\boldsymbol{x}} + \frac{A}{\eta_1} \leq 10GDA.
$$

Therefore, $\tilde{R}_T^{\boldsymbol{x}} = O(\sqrt{W_T^{\boldsymbol{x}} \log T})$ holds.

We can get $O(\sqrt{V_T^{\boldsymbol{x}} d \log T})$ bound as follows:

$$
\begin{aligned}
R_T^{\boldsymbol{x}} \leq \tilde{R}_T^{\boldsymbol{x}} &= \frac{1}{\eta} \sum_{t=1}^{T} (\eta^2 (\langle \nabla f_t(\boldsymbol{x}_t), \boldsymbol{x}_t - \boldsymbol{x} \rangle)^2 - \ell_t^{\eta}(\boldsymbol{x})) \\
&= \eta \sum_{t=1}^{T} (\langle \nabla f_t(\boldsymbol{x}_t), \boldsymbol{x}_t - \boldsymbol{x} \rangle)^2 + \frac{1}{\eta} \left( \sum_{t=1}^{T} (\ell_t(\boldsymbol{x}_t) - \ell_t(\boldsymbol{x}_t^{\ell})) + \sum_{t=1}^{T} (\ell_t(\boldsymbol{x}_t^{\ell}) - \ell_t(\boldsymbol{x})) \right) \\
&\leq \eta V_T^{\boldsymbol{x}} + \frac{B}{\eta},
\end{aligned}
$$

where $B := 2\log\left(\sqrt{3}((1/2)\log_2 T + 3)\right) + 10d \log T$ and the last inequality follows from inequalities (16) and (19). By similar arguments, $\tilde{R}_T^{\boldsymbol{x}} = O(\sqrt{V_T^{\boldsymbol{x}} d \log T})$ holds. $\qquad \square$

### B.6 Proof of Lemma 5.4

In this subsection, we prove Lemma 5.4 used in the proof of Theorem 5.3.

*Proof.* Since $x \leq 3(a+b)/2$ holds when $x \leq b$, it is sufficient to consider the case $x > b$. If $x \leq \sqrt{ax} + b$, then we have

$$
x^2 - (a + 2b)x + b^2 \leq 0.
$$

By solving this, we have

$$
x \leq \frac{a + 2b + \sqrt{a^2 + 4ab}}{2} \leq a + b + \sqrt{ab} \leq \frac{3}{2}(a + b).
$$

The second inequality holds from the inequality $\sqrt{x+y} \leq \sqrt{x} + \sqrt{y}$ for $x, y \geq 0$, and the last inequality holds from the inequality of arithmetic and geometric means. $\qquad \square$

## B.7 Proof of Corollary 5.5 for USC

This subsection presents the proof of Corollary 5.5 for USC.

*Proof.* The regret satisfies

$$R_T = \sum_{t=1}^{T} f_t(\boldsymbol{x}_t) - \sum_{t=1}^{T} f_t(\boldsymbol{x}_t^i) + \sum_{t=1}^{T} f_t(\boldsymbol{x}_t^i) - \min_{\boldsymbol{x} \in \mathcal{X}} \sum_{t=1}^{T} f_t(\boldsymbol{x})$$
$$= R_T^{\mathrm{meta}} + R_T^{\mathrm{expert}},$$

where $R_T^{\mathrm{meta}} := \sum_{t=1}^{T} f_t(\boldsymbol{x}_t) - \sum_{t=1}^{T} f_t(\boldsymbol{x}_t^i)$ and $R_T^{\mathrm{expert}} := \sum_{t=1}^{T} f_t(\boldsymbol{x}_t^i) - \min_{\boldsymbol{x} \in \mathcal{X}} \sum_{t=1}^{T} f_t(\boldsymbol{x})$. From Theorem A.1,

$$R_T^{\mathrm{meta}} \leq \sum_{t=1}^{T} \langle \nabla f_t(\boldsymbol{x}_t), \boldsymbol{x}_t - \boldsymbol{x}_t^i \rangle$$

$$\leq 4\Gamma G D + \frac{\Gamma}{\sqrt{\log |\mathcal{E}|}} \sqrt{4G^2 D^2 + \sum_{t=1}^{T} (\langle \nabla f_t(\boldsymbol{x}_t), \boldsymbol{x}_t - \boldsymbol{x}_t^i \rangle)^2}$$

$$\leq 2\Gamma G D \left( 2 + \frac{1}{\sqrt{\log |\mathcal{E}|}} \right) + \sqrt{\frac{\Gamma^2}{\log |\mathcal{E}|} \sum_{t=1}^{T} (\langle \nabla f_t(\boldsymbol{x}_t), \boldsymbol{x}_t - \boldsymbol{x}_t^i \rangle)^2}.$$

Last inequality holds from the inequality $\sqrt{x + y} \leq \sqrt{x} + \sqrt{y}$ for $x, y \geq 0$.

Similar to equation (2), if $R_T^{\mathrm{meta}} > 0$, inequality

$$\sum_{t=1}^{T} (\langle \nabla f_t(\boldsymbol{x}_t), \boldsymbol{x}_t - \boldsymbol{x}_t^i \rangle)^2 \leq \frac{2}{\gamma} \sum_{t=1}^{T} \langle \nabla f_t(\boldsymbol{x}_t), \boldsymbol{x}_t - \boldsymbol{x}_t^i \rangle + k_\gamma G^2 D^2$$

holds. By combining these inequalities, we have

$$\sum_{t=1}^{T} \langle \nabla f_t(\boldsymbol{x}_t), \boldsymbol{x}_t - \boldsymbol{x}_t^i \rangle \leq 2\Gamma G D \left( 2 + \frac{1}{\sqrt{\log |\mathcal{E}|}} \right) + \sqrt{\frac{\Gamma^2}{\log |\mathcal{E}|} \left( \frac{2}{\gamma} \sum_{t=1}^{T} \langle \nabla f_t(\boldsymbol{x}_t), \boldsymbol{x}_t - \boldsymbol{x}_t^i \rangle + k_\gamma G^2 D^2 \right)}$$

$$\leq \Gamma G D \left( 4 + \frac{2 + \sqrt{k_\gamma}}{\sqrt{\log |\mathcal{E}|}} \right) + \sqrt{\frac{2\Gamma^2}{\gamma \log |\mathcal{E}|} \sum_{t=1}^{T} \langle \nabla f_t(\boldsymbol{x}_t), \boldsymbol{x}_t - \boldsymbol{x}_t^i \rangle}.$$

From Lemma 5.4 with $a = 2\Gamma^2/(\gamma \log |\mathcal{E}|)$ and $b = \Gamma G D(4 + (2 + \sqrt{k_\gamma})/\sqrt{\log |\mathcal{E}|})$, we have

$$R_T^{\mathrm{meta}} \leq \sum_{t=1}^{T} \langle \nabla f_t(\boldsymbol{x}_t), \boldsymbol{x}_t - \boldsymbol{x}_t^i \rangle$$

$$\leq \frac{3}{2} \left( \Gamma G D \left( 4 + \frac{2 + \sqrt{k_\gamma}}{\sqrt{\log |\mathcal{E}|}} \right) + \frac{2\Gamma^2}{\gamma \log |\mathcal{E}|} \right).$$

Since $|\mathcal{E}| = O(\log T)$ in USC, we obtain the following loose upper bound:

$$R_T^{\mathrm{meta}} = O \left( \frac{d}{\gamma} \log T + G D \sqrt{kd \log T} \right).$$

On the other hand, by thinking of the case that $i$th expert is MetaGrad or Maler, from Corollary 5.5 for MetaGrad and Maler,

$$R_T^{\mathrm{expert}} = O \left( \frac{d}{\gamma} \log T + G D \sqrt{kd \log T} \right).$$

Combining these bounds, we get

$$R_T = O\left(\frac{d}{\gamma}\log T + GD\sqrt{kd\log T}\right).$$

$\square$

### B.8 Proof of Theorem 5.6

This subsection presents the proof of Theorem 5.6.

*Proof.* From the definition of strong convexity, we have

$$
\begin{aligned}
R_T^{\boldsymbol{x}} &= \sum_{t=1}^{T}(f_t(\boldsymbol{x}_t) - f_t(\boldsymbol{x})) \\
&\leq \sum_{t=1}^{T}\left(\langle \nabla f_t(\boldsymbol{x}_t), \boldsymbol{x}_t - \boldsymbol{x}\rangle - \frac{\lambda_t}{2}\|\boldsymbol{x}_t - \boldsymbol{x}\|^2\right) \\
&= \tilde{R}_T^{\boldsymbol{x}} - \frac{\lambda}{2G^2}W_T^{\boldsymbol{x}} + \sum_{t=1}^{T}\frac{\lambda - \lambda_t}{2}\|\boldsymbol{x}_t - \boldsymbol{x}\|^2 \\
&\leq \tilde{R}_T^{\boldsymbol{x}} - \frac{\lambda}{2G^2}W_T^{\boldsymbol{x}} + \sum_{t:\lambda_t<\lambda}\frac{\lambda - \lambda_t}{2}D^2 \\
&\leq \tilde{R}_T^{\boldsymbol{x}} - \frac{\lambda}{2G^2}W_T^{\boldsymbol{x}} + \frac{\lambda}{2}k_\lambda D^2.
\end{aligned}
$$

If $R_T^{\boldsymbol{x}} < 0$, 0 is the upper bound, so it is sufficient to think of the case $R_T^{\boldsymbol{x}} \geq 0$. In this case, we have

$$W_T^{\boldsymbol{x}} \leq \frac{2G^2}{\lambda}\tilde{R}_T^{\boldsymbol{x}} + k_\lambda G^2 D^2.$$

From the assumption of Theorem 5.6, there exists a positive constant $C > 0$ such that

$$
\begin{aligned}
\tilde{R}_T^{\boldsymbol{x}} &\leq C\left(\sqrt{W_T^{\boldsymbol{x}}r_1(T)} + r_2(T)\right) \\
&\leq C\left(\sqrt{\left(\frac{2G^2}{\lambda}\tilde{R}_T^{\boldsymbol{x}} + k_\lambda G^2 D^2\right)r_1(T)} + r_2(T)\right) \\
&\leq \sqrt{\frac{2G^2}{\lambda}C^2 r_1(T)\tilde{R}_T^{\boldsymbol{x}}} + CGD\sqrt{k_\lambda r_1(T)} + Cr_2(T).
\end{aligned}
$$

Last inequality holds from the inequality $\sqrt{x+y} \leq \sqrt{x} + \sqrt{y}$ for $x, y \geq 0$. Here, we use Lemma 5.4 with $a = (2G^2/\lambda)C^2 r_1(T)$ and $b = CGD\sqrt{k_\lambda r_1(T)} + Cr_2(T)$,

$$\tilde{R}_T^{\boldsymbol{x}} \leq \frac{3}{2}\left(\frac{2G^2}{\lambda}C^2 r_1(T) + CGD\sqrt{k_\lambda r_1(T)} + Cr_2(T)\right).$$

From this inequality and $R_T^{\boldsymbol{x}} \leq \tilde{R}_T^{\boldsymbol{x}}$, Theorem 5.6 follows. $\square$

### B.9 Proof of Corollary 5.7 for MetaGrad and Maler

This subsection presents the proof of Corollary 5.7 for MetaGrad and Maler.

*Proof.* As for MetaGrad and Maler, from Theorem 5.1 and Theorem 5.2,

$$\tilde{R}_T^{\boldsymbol{x}} = O(\sqrt{W_T^{\boldsymbol{x}}\tilde{d}\log T} + GD\tilde{d}\log T)$$

holds for any $\boldsymbol{x} \in \mathcal{X}$, where $\tilde{d}$ is $d$ and 1 in the case of MetaGrad and Maler, respectively. Therefore, by Theorem 5.6, we have

$$R_T^{\boldsymbol{x}} = O\left(\left(\frac{G^2}{\lambda} + GD\right)\tilde{d}\log T + GD\sqrt{k_\lambda \tilde{d}\log T}\right).$$

Here, $k_\lambda$ satisfies

$$k_\lambda = \sum_{t=1}^T \max\left\{1 - \frac{\lambda_t}{\lambda}, 0\right\} = \sum_{t:\,\lambda_t < \lambda}\left(1 - \frac{\lambda_t}{\lambda}\right) \leq k.$$

Hence, we have

$$R_T^{\boldsymbol{x}} = O\left(\left(\frac{G^2}{\lambda} + GD\right)\tilde{d}\log T + GD\sqrt{k\tilde{d}\log T}\right),$$

and especially,

$$R_T = O\left(\left(\frac{G^2}{\lambda} + GD\right)\tilde{d}\log T + GD\sqrt{k\tilde{d}\log T}\right).$$

$\square$

## B.10 Proof of Corollary 5.7 for USC

This subsection presents the proof of Corollary 5.7 for USC.

*Proof.* The same as the proof of Corollary 5.5, we have

$$R_T = R_T^{\text{meta}} + R_T^{\text{expert}}.$$

From Theorem A.1, we have

$$R_T^{\text{meta}} \leq \sum_{t=1}^T \left\langle \nabla f_t(\boldsymbol{x}_t), \boldsymbol{x}_t - \boldsymbol{x}_t^i \right\rangle$$

$$\leq 2\Gamma GD\left(2 + \frac{1}{\sqrt{\log|\mathcal{E}|}}\right) + \sqrt{\frac{\Gamma^2}{\log|\mathcal{E}|}\sum_{t=1}^T \left\langle \nabla f_t(\boldsymbol{x}_t), \boldsymbol{x}_t - \boldsymbol{x}_t^i \right\rangle^2}$$

$$\leq 2\Gamma GD\left(2 + \frac{1}{\sqrt{\log|\mathcal{E}|}}\right) + \sqrt{\frac{\Gamma^2 G^2}{\log|\mathcal{E}|}\sum_{t=1}^T \|\boldsymbol{x}_t - \boldsymbol{x}_t^i\|^2}.$$

From the definition of strong convexity, we have

$$R_T^{\text{meta}} = \sum_{t=1}^T (f_t(\boldsymbol{x}_t) - f_t(\boldsymbol{x}_t^i))$$

$$\leq \sum_{t=1}^T \left(\left\langle \nabla f_t(\boldsymbol{x}_t), \boldsymbol{x}_t - \boldsymbol{x}_t^i \right\rangle - \frac{\lambda}{2}\|\boldsymbol{x}_t - \boldsymbol{x}_t^i\|^2\right) + \sum_{t=1}^T \frac{\lambda}{2}\max\left\{1 - \frac{\lambda_t}{\lambda}, 0\right\}\|\boldsymbol{x}_t - \boldsymbol{x}_t^i\|^2$$

$$\leq \sum_{t=1}^T \left\langle \nabla f_t(\boldsymbol{x}_t), \boldsymbol{x}_t - \boldsymbol{x}_t^i \right\rangle - \frac{\lambda}{2}\sum_{t=1}^T \|\boldsymbol{x}_t - \boldsymbol{x}_t^i\|^2 + \frac{\lambda}{2}k_\lambda D^2.$$

If $R_T^{\text{meta}} < 0$, 0 is the upper bound, so it is sufficient to think of the case $R_T^{\text{meta}} \geq 0$. In this case, we have

$$\sum_{t=1}^T \|\boldsymbol{x}_t - \boldsymbol{x}_t^i\|^2 \leq \frac{2}{\lambda}\sum_{t=1}^T \left\langle \nabla f_t(\boldsymbol{x}_t), \boldsymbol{x}_t - \boldsymbol{x}_t^i \right\rangle + k_\lambda D^2.$$

By combining these inequalities, we have

$$\sum_{t=1}^{T} \langle \nabla f_t(\boldsymbol{x}_t), \boldsymbol{x}_t - \boldsymbol{x}_t^i \rangle \leq 2\Gamma GD \left( 2 + \frac{1}{\sqrt{\log |\mathcal{E}|}} \right) + \sqrt{\frac{\Gamma^2 G^2}{\log |\mathcal{E}|} \left( \frac{2}{\lambda} \sum_{t=1}^{T} \langle \nabla f_t(\boldsymbol{x}_t), \boldsymbol{x}_t - \boldsymbol{x}_t^i \rangle + k_\lambda D^2 \right)}$$

$$\leq \Gamma GD \left( 4 + \frac{2 + \sqrt{k_\lambda}}{\sqrt{\log |\mathcal{E}|}} \right) + \sqrt{\frac{2\Gamma^2 G^2}{\lambda \log |\mathcal{E}|} \sum_{t=1}^{T} \langle \nabla f_t(\boldsymbol{x}_t), \boldsymbol{x}_t - \boldsymbol{x}_t^i \rangle}.$$

From Lemma 5.4 with $a = 2\Gamma^2 G^2/(\lambda \log |\mathcal{E}|)$ and $b = \Gamma GD(4 + (2 + \sqrt{k_\lambda})/\sqrt{\log |\mathcal{E}|})$, we have

$$R_T^{\text{meta}} \leq \sum_{t=1}^{T} \langle \nabla f_t(\boldsymbol{x}_t), \boldsymbol{x}_t - \boldsymbol{x}_t^i \rangle \leq \frac{3}{2} \left( \Gamma GD \left( 4 + \frac{2 + \sqrt{k_\lambda}}{\sqrt{\log |\mathcal{E}|}} \right) + \frac{2\Gamma^2 G^2}{\lambda \log |\mathcal{E}|} \right).$$

Since $|\mathcal{E}| = O(\log T)$ in USC, we obtain the following loose upper bound:

$$R_T^{\text{meta}} = O\left( \left( \frac{G^2}{\lambda} + GD \right) \log T + GD\sqrt{k \log T} \right).$$

On the other hand, by thinking of the case that $i$th expert is Maler, from Corollary 5.7 for Maler, we have

$$R_T^{\text{expert}} = O\left( \left( \frac{G^2}{\lambda} + GD \right) \log T + GD\sqrt{k \log T} \right).$$

Combining these bounds, we get

$$R_T = O\left( \left( \frac{G^2}{\lambda} + GD \right) \log T + GD\sqrt{k \log T} \right).$$

$\square$

### B.11   The Lemma in the Proof of Theorem 6.2

In this subsection, we introduce the following lemma used in the proof of Theorem 6.2.

**Lemma B.5.** *(Hazan, 2016) Let $\boldsymbol{A} \succeq \boldsymbol{B} \succ \boldsymbol{O}$ be positive definite matrices. We then have*

$$\text{tr}\big(\boldsymbol{A}^{-1}(\boldsymbol{A} - \boldsymbol{B})\big) \leq \log \frac{|\boldsymbol{A}|}{|\boldsymbol{B}|}.$$

## C   Numerical Experiments

In this section, we explain experimental results. We compare the performances of 5 OCO algorithms; OGD with stepsizes $\eta_t = D/(G\sqrt{t})$, ONS, MetaGrad, Algorithm 1 with $S_1 = \emptyset$ (Con-ONS), and Algorithm 1 with $S_2 = \emptyset$ (Con-OGD). We include OGD, ONS, and MetaGrad because OGD and ONS are famous OCO algorithms, and MetaGrad is one of the universal algorithms. All the experiments are implemented in Python 3.9.2 on a MacBook Air whose processor is 1.8 GHz dual-core Intel Core i5 and memory is 8GB.

### C.1   Experiment 1: Contaminated Exp-Concave

In this experiment, we set $d = 1$, $\mathcal{X} = [0, 1]$ and the objective function is as follows:

$$f_t(x) := \begin{cases} 100x & t \in I, \\ -\log(x + 0.01) & \text{otherwise}, \end{cases}$$

where $I \subset [T]$ is chosen uniformly at random under the condition that $|I| = k$. $(f_1, f_2, \ldots, f_T)$ is $k$-contaminated 1-exp-concave and the minimum value of $\sum_{t=1}^{T} f_t$ is $T - 2k - (T - k) \log((T - k)/(100k))$ if

Table 2: Parameter setting in Experiment 1.

| $x_1$ | $x^*$ | $T$ | $k$ | $D$ | $G$ | $\alpha$ |
|---|---|---|---|---|---|---|
| 0 | 0.02 | 1000 | 250 | 1 | 100 | 1 |

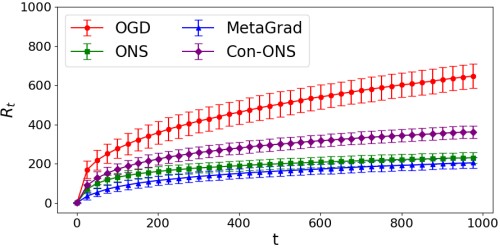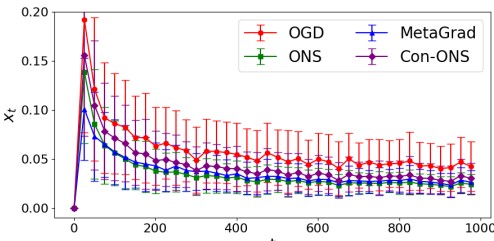

Figure 1: The comparison of the time variation of regret (left) and $x_t$ (right) in Experiment 1.

$2k < T$. We repeat this numerical experiment in 100 different random seeds and calculate their mean and standard deviation. Other parameters are shown in Table 2.

We compare the performances of 4 OCO algorithms: OGD, ONS, MetaGrad, and Con-ONS. The parameters of ONS are set as $\gamma = 0.005$ and $\varepsilon = 1/(\gamma^2 D^2) = 40000$.

The time variation of regret and $x_t$ is shown in Figure 1. In the graphs presented in this paper, the error bars represent the magnitude of the standard deviation. Standard deviations in the graphs are large because contamination causes fluctuation in the sequence of solutions. Only points where t is a multiple of 25 are plotted to view the graph easily. The left graph shows that the regrets of all methods are sublinear. MetaGrad and ONS perform particularly well, followed by Con-ONS. From the right graph, we can confirm that $x_t$ of all methods converge to the optimal solution quickly. OGD seems influenced by contamination a little stronger than other methods.

## C.2 Experiment 2: Contaminated Strongly Convex

In this experiment, $d = 1$, $\mathcal{X} = [0, 1]$ and the objective function is as follows:

$$f_t(x) := \begin{cases} x & t \in I, \\ \frac{1}{2}(x - 1)^2 & \text{otherwise,} \end{cases}$$

where $I \subset [T]$ is chosen uniformly at random under the condition that $|I| = k$. $(f_1, f_2, \ldots, f_T)$ is $k$-contaminated 1-strongly convex and the minimum value of $\sum_{t=1}^{T} f_t$ is $(2kT - 3k^2)/(2T - 2k)$ if $2k < T$. We repeat this numerical experiment in 100 different random seeds and calculate their mean and standard deviation. Other parameters are shown in Table 3.

We compare the performances of 3 OCO algorithms: OGD, MetaGrad, and Con-OGD.

The time variation of regret and $x_t$ is shown in Figure 2. Only points where $t$ is a multiple of 25 are plotted to view the graph easily. The left graph shows that the regrets of all methods are sublinear. The performance of

Table 3: Parameter setting in Experiment 2.

| $x_1$ | $x^*$ | $T$ | $k$ | $D$ | $G$ | $\lambda$ |
|---|---|---|---|---|---|---|
| 0 | 2/3 | 1000 | 250 | 1 | 1 | 1 |

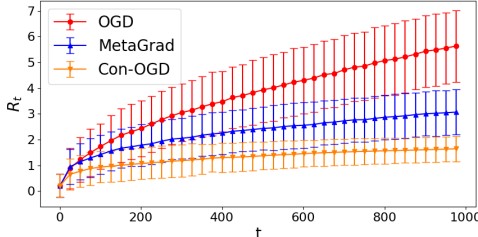 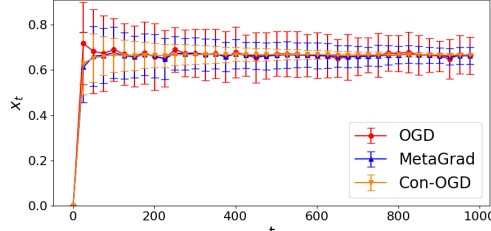

Figure 2: The comparison of the time variation of regret (left) and $x_t$ (right) in Experiment 2.

Table 4: Parameter setting in Experiment 3.

| $\boldsymbol{x}_1$ | $n$ | $d$ | $T$ | $k$ | $D$ |
|---|---|---|---|---|---|
| $\mathbf{0}$ | 10 | 5 | 1000 | 250 | $\sqrt{5}$ |

Con-OGD is the best, followed by MetaGrad and OGD, showing correspondence with the theoretical results. From the right graph, we can confirm that $x_t$ of all methods converge to the optimal solution quickly.

### C.3 Experiment 3: Mini-Batch Least Mean Square Regressions

Experimental settings are as follows. We use the squared loss as the objective function:

$$f_t(\boldsymbol{x}) := \frac{1}{n} \sum_{i=1}^n (\langle \boldsymbol{a}_{t,i}, \boldsymbol{x} \rangle - b_{t,i})^2,$$

which is exemplified in Example 3.7. In this experiment, each component of the vector $\boldsymbol{a}_{t,i}$ is taken from a uniform distribution on $[1, 2]$ independently. We set $\mathcal{X} = [0, 1]^d$ and assume there exists an optimal solution $\boldsymbol{x}^*$ which is taken from a uniform distribution on $\mathcal{X}$, i.e., we take $b_{t,i} = \langle \boldsymbol{a}_{t,i}, \boldsymbol{x}^* \rangle$. We set $k$ firstly and compute thresholds $\alpha$ and $\lambda$ based on $k$, though this is impossible in real applications. Parameters $G$, $\lambda$, $\alpha$ are calculated for each $\boldsymbol{a}_{t,i}$ and $b_{t,i}$, e.g., $G \simeq 429$, $\lambda \simeq 0.0969$, and $\alpha \simeq 5.28 \times 10^{-7}$ for some sequence. The parameters of ONS are set as described in Section A.2. Other parameters are shown in Table 4.

The time variation of regret and $\|\boldsymbol{x}_t\|$ is shown in Figure 3. The performances of OGD, Con-OGD, and Con-ONS are almost the same in this experiment. The left graph shows that OGD's, MetaGrad's, and our proposed method's regrets are sublinear and consistent with the theoretical results. Though this is out of the graph, ONS's regret is almost linear even if we take $T = 10000$. From the right graph, we can confirm that $\|\boldsymbol{x}_t\|$ of OGD, MetaGrad, and proposed methods converge to some point quickly, and that of ONS does not change so much. The poor performance of ONS is because $\gamma$ is too small to take large enough stepsizes. This result shows the vulnerability of ONS.

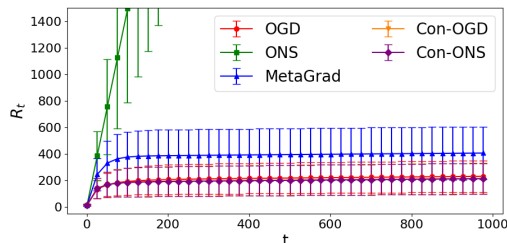 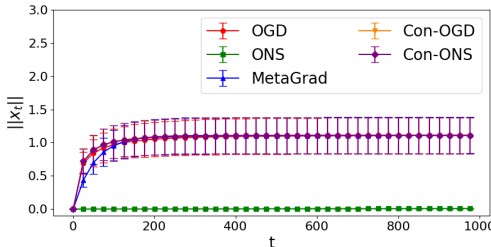

Figure 3: The comparison of the time variation of regret (left) and $\|\boldsymbol{x}_t\|$ (right).

