# OpenReview forum: "Contaminated Online Convex Optimization"
_TMLR — Accepted by TMLR_

### Review · Reviewer_5RWn · 2024-09-08

**Summary Of Contributions:**

In the standard OCO setting, it is generally assumed that the losses have the same base properties on all rounds (the losses are all strongly convex, exp-concave, smooth, Lipschitz, etc). This paper investigates the setting where k of the T losses are "contaminated" and do not share the same properties as the rest. In this setting, the paper provides a regret lower bound
of $\Omega(\log(T) + \sqrt{k})$ and algorithms achieving a nearly matching upper bound of $O(\log(T) + \sqrt{k\log(T)})$, for both contaminated strongly convex losses and for contaminated exp-concave losses.
It is also shown that a regret bound of $O(\log(T)+\sqrt{k})$ can be attained when given the additional information of which type of loss was encountered in each round.

**Audience:**

Yes

**Claims And Evidence:**

Yes

**Requested Changes:**

My only request for changes is to discuss the issue with the lower bound, discussed in the weaknesses section above.

See strengths/weaknesses for suggestions that I think would improve the clarity of the paper.

**Strengths And Weaknesses:**

## Strengths

The paper studies an interesting generalization of the standard OCO setting and provides reasonable examples where such a setting would
naturally be encountered. The paper is easy to read throughout and the proofs easy to follow. The paper provides lower bounds for the new setting, as well as algorithms achieving nearly-matching regret upper bounds and matching upper bounds (provided additional information).

## Weaknesses

- The lower bound is not necessarily tight
  - The result in section 6 is (seemingly) meant to provide a result that matches the $\Omega(\log(T)+\sqrt{k})$ lower bound derived earlier, indicating that the lower bound is tight. However, the setting with extra information is a strictly easier problem setting, so this result does not actually prove that the lower bound is tight: it could still be the case that the $O(\log(T)+\sqrt{k\log(T)})$ upper bound is tight and the lower bound is loose and should have $\sqrt{k\log(T)}$ rather than just $\sqrt{k}$.

- It is rather annoying that Theorem 5.2 is derived for $x^\*$, and the reader just has to believe that it holds for arbitrary comparators $x$. I believe this is true, but it feels too hand-wavy for the paper to rely on a result that hasn't been formally proven anywhere. Perhaps Theorem 5.2 could be proven for completeness in an appendix.


- Experiments:
  - all the error bars for ONS and the proposed algorithms seem to overlap --- this suggests there is no statistically significant difference between the performance of the proposed algorithm and naively applying ONS.

- Questions:
  - The setting described here still has losses with the same property on all rounds except for the contaminated ones. Might it more generally be possible to let the loss type vary on each round --- being either strongly-convex, exp-concave, general convex, etc --- by developing a strongly-adaptive universal algorithm? Then for any partition of $[1,T]$, where the losses are of the same type on each interval, we would be able to make the correct guarantee on that sub-interval. This would be a strictly more general setting and more general result. Do you expect such a result to be achievable?
  - Theorem 6.2: why restrict to comparing against the argmin $x^\*$? If the results holds for an arbitrary comparator you should state it that way, otherwise the next paper will again have to state "their analysis applies not just for $x^\*$", as had to be done in this paper when discussing the algorithm of Wang et al. (2020).

- Suggestions (not requests) which I think could improve the clarity of this work
  - Overall structure:
    - It feels like sections 7 and 4 should be joined into a single section. That is, provide the lower bound of Theorem 4.2 and
      its two corollaries from section 7, giving the lower bounds for this new problem setting. This is the more important result as it shows
      what the optimal result is that any algorithm should be shooting for in this contaminated OCO setting. Then, the
      vulnerability of ONS can be shown as a special case, as a subsection, demonstrating why the naive approach is insufficient. I think this would be a cleaner layout overall.
  - Section 5:
    - I think the core argument is easier to see by just considering the two cases, rather than writing the
      $\frac{\gamma}{2}\max\{1-\gamma_t/\gamma,0\}$, which is a bit puzzling to understand. The expression with the max is obviously more concise, but takes more effort for the reader to understand. Given that this proof is given in the main text, it would be ideal for it to be as simple as possible.

      In particular, we always have
      \begin{align*}
          \sum_t \ell_t(w_t)-\ell_t(u)
          &\le
              \tilde R_T(u) - \sum_t \frac{\alpha_t}{2}\langle g_t, w_t-u\rangle^2\\\\
          &=
              \tilde R_T(u) - \sum_{t:\alpha_t\ge \gamma}\frac{\alpha_t}{2}\langle g_t, w_t-u\rangle ^2-\sum_{t:\alpha_t<\gamma}\frac{\alpha_t}{2}\langle g_t, w_t-u\rangle^2\\\\
          &\le
              \tilde R_T(u) - \frac{\gamma}{2}V_T-\sum_{t:\alpha_t<\gamma}\frac{\alpha_t}{2}\langle g_t, w_t-u\rangle^2.
      \end{align*}
      On the $k_\gamma$ rounds where $\alpha_t<\gamma$,
      we can still expose a $-\frac{\gamma}{2}\langle g_t,w_t-u\rangle^2$ term by adding and subtracting:
      \begin{align*}
      -\frac{\alpha_t}{2}\langle g_t,w_t-u\rangle^2= -\frac{\gamma}{2}\langle g_t,w_t-u\rangle ^2 + \frac{1}{2}\left(\gamma-\alpha_t\right)\langle g_t,w_t-u\rangle^2\le -\frac{\gamma}{2}\langle g_t,w_t-u\rangle^2 + \frac{\gamma}{2}G^2D^2,
      \end{align*}
      so overall we have
      \begin{align*}
      \sum_t \ell_t(w_t)-\ell_t(u) \le \tilde R_T(u) - \frac{\gamma}{2}V_T + \frac{\gamma}{2}k_\gamma G^2D^2.
      \end{align*}


- Minor details:
  - p. 11: "low of cosines"

---

> ### Author Response · Authors · 2024-10-10
>
> We appreciate your thorough review and valuable comments.
> We hope the following replies address your concerns and questions:
>
> ## Weaknesses
>
> > The lower bound is not necessarily tight. The result in section 6 is (seemingly) meant to provide a result that matches the $\Omega(\log(T)+\sqrt{k})$ lower bound derived earlier, indicating that the lower bound is tight. However, the setting with extra information is a strictly easier problem setting, so this result does not actually prove that the lower bound is tight: it could still be the case that the $O(\log(T)+\sqrt{k\log(T)})$ upper bound is tight and the lower bound is loose and should have $\sqrt{k\log(T)}$ rather than just $\sqrt{k}$.
>
> We agree with your point and acknowledge that there is a gap between the upper and lower bounds.
> We conjecture that the upper bound could potentially be improved.
> To avoid any misleading expressions, we have revised the text (e.g., we have removed the sentence "However, the regret analyses in this section are insightful because the upper bound with additional $O(k)$ term implies that the regret upper bounds of universal algorithms in Section 5 can be improved." in Remark 6.4 (p.12)).
>
> > It is rather annoying that Theorem 5.2 is derived for $x^*$, and the reader just has to believe that it holds for arbitrary comparators $x$. I believe this is true, but it feels too hand-wavy for the paper to rely on a result that hasn't been formally proven anywhere. Perhaps Theorem 5.2 could be proven for completeness in an appendix.
>
> Thank you for pointing this out.
> We agree that a formal proof would improve the completeness of the paper.
> We have added the proof of Theorem 5.2 in Appendix B.5 (pp.21--23).
>
> > all the error bars for ONS and the proposed algorithms seem to overlap --- this suggests there is no statistically significant difference between the performance of the proposed algorithm and naively applying ONS.
>
> We re-ran the experiments with a larger $T$ to address this concern.
> The new figures show fewer overlaps, though some overlap persists.
> Unfortunately, the results still do not demonstrate a clear performance superiority of our proposed methods.
> We regard these results as supplemental.
>
> Additionally, during the re-experiments, we found that the poor performance of OGD was due to the stepsize not being optimal.
> OGD with an optimal stepsize performs better, particularly in Experiment 3.
>
> ## Questions
>
> > The setting described here still has losses with the same property on all rounds except for the contaminated ones. Might it more generally be possible to let the loss type vary on each round --- being either strongly-convex, exp-concave, general convex, etc --- by developing a strongly-adaptive universal algorithm? Then for any partition of $[1,T]$, where the losses are of the same type on each interval, we would be able to make the correct guarantee on that sub-interval. This would be a strictly more general setting and more general result. Do you expect such a result to be achievable?
>
> We do not yet have a concrete approach to this broader setting.
> However, we agree that it is an interesting and potentially achievable research direction.
> In cases where function class information is available, Algorithm 1 already performs as a strongly-adaptive universal algorithm.
> However, if only gradient information is accessible, the main challenge lies in detecting and adapting to the changing function classes.
>
> > Theorem 6.2: why restrict to comparing against the argmin $x^*$? If the results holds for an arbitrary comparator you should state it that way, otherwise the next paper will again have to state "their analysis applies not just for $x^*$", as had to be done in this paper when discussing the algorithm of Wang et al. (2020).
>
> The upper bound of Theorem 6.2 does not depend on $x^\ast$ (c.f. Theorem 5.2), and $R_T^x\leq R_T$ holds.
> Therefore, it is sufficient to discuss it only for $x^\ast$.

---

> > ### Comment · Reviewer_5RWn · 2024-10-26
> >
> > Thank you for addressing my concerns in the main text. I would be happy to recommend acceptance now that these details have been made clear in the main text

---

### Review · Reviewer_H5j2 · 2024-09-25

**Summary Of Contributions:**

The paper deals with contaminated OCO; an intermediate regime between general OCO, where no additional structural assumptions are made on the sequence of convex functions $f_{1},\dots,f_{T}$ and $\mathcal{F}$-OCO, whereby all such functions are assumed to be in a function class $\mathcal{F}$, where curvature may be leveraged to achieve fast $O(\log(T))$ rates. The contamination $k$ is defined as the number of rounds whereby general OCO is the correct abstraction, and no other additional structure can be usefully exploited.

The paper provides a general algorithm to deal with arbitrary mixtures of exp-concave, strongly convex and losses from the latter contaminated class. A simple example of this being mini-batch least-square regression, with a corruption defined as the rounds for which the outer product of the sum of features has eigenvalue below the threshold used to tune the algorithm.

The authors begin by providing a lower bound for any online convex optimisation algorithm demonstrating that the worst-case regret for any OCO algorithm can be decoupled into two terms $g_{1}(T)$-describing the regret for structured losses and $g_{2}(T)$ for corrupted ones, which are additive in the lower bound (i.e. $R_{T}\geq g_{1}(T)+g_{2}(k)$). With some extra work, the inadequacy of Online Newton Step for sufficiently corrupted sequences is proven in a $\Omega(T)$ lower bound. It is then demonstrated that some existing "universal" algorithms (those which adapt to a _fixed_ function class, without prior knowledge in advance) for OCO (MetaGrad, Maler and USC) are also adaptive in the corrupted case, albeit with an $O(\sqrt{\log(T)})$ sub-optimality with respect to Algorithm 1, which leverages additional side information on the function class of the previous objective function to be revealed. Experiments are included to verify theoretical conclusions.

**Audience:**

Yes

**Broader Impact Concerns:**

There are no ethical concerns.

**Claims And Evidence:**

Yes

**Requested Changes:**

I have no requested adjustments to this paper. Broadly speaking, results are clear and the paper is well structured; I do not think too much should be added that wouldn't constitute another study altogether.

**Strengths And Weaknesses:**

Strengths of the paper:
- The topic of the paper is a very natural question to ask in the context of online convex optimisation.
- The paper contains a wealth of interesting theoretical results characterising algorithms for the setting introduced.
- The paper is well structured, and many natural side considerations have been addressed by way of complementary remarks throughout.
- Results appear to be correct.

Weaknesses:
- While the analysis is more detailed, the only algorithmic contribution is Algorithm 1 - constituting a slight generalisation of ONS/OCO for strongly convex losses.
- The algorithm provided requires the function class of the previous objective to be revealed to the algorithm in contrast to the three other comparison algorithms, all of which only require gradient information. This leaves a gap in practicality of Algorithm 1 with respect to the others. The authors have been clear about this discrepancy.
- As a very mild addition to this list - there does remain an $O(\sqrt{\log(T)})$ sub-optimality between the upper and lower bounds, an explanation of which would be extremely interesting. Considering that it turns up in the corruption term in all cases where only gradients (but not the class) of the current iterate is available, an explanation of this would significantly add to our understanding of the interplay between corruption and information passed to the learner. Having said that, this is fine to be left for future work considering the contributions present. Arguably, this paper's having discovered this gap in the first place is a strength.

---

> ### Author Response · Authors · 2024-10-10
>
> We sincerely appreciate your positive feedback and encouragement.
> As you noted, further research in this regime will be necessary to enhance the practicality of our proposed methods.

---

### Review · Reviewer_vD1L · 2024-09-26

**Summary Of Contributions:**

The paper considers the online convex optimization problem, where loss functions may vary between being strongly convex, exp-concave, or simply convex at different time steps. This setting, termed contaminated online convex optimization, introduces greater complexity. The authors establish lower bounds for this problem and demonstrate that the popular Online Newton Step (ONS) algorithm is vulnerable in such a setting. They also show that current adaptive algorithms, based on the analysis in the paper, do not yet reach these bounds. However, they show that with additional information, they can derive an algorithm that achieves the lower bound.

**Audience:**

Yes

**Broader Impact Concerns:**

It's fundamental work in online convex optimization, and I don't see direct ethical implications for the work.

**Claims And Evidence:**

Yes

**Requested Changes:**

# Main

Please refer to the comments in the weakness part as well as the comments for example 3.7, 3.8, Remark 3.9 and Appendix C.3.

# Minor

- As you also provide proof of exp-concavity of functions in Example 3.8, it is worth anyways mentioning that in the body.

- There is already the dimension d in Table 1 but it was only mentioned later.

- In the proof of Theorem 5.3, you discussed two cases where R_T^x<0 and R_T^x \ge 0. In this case, \tilde R_T^x can still be 0. You might want to change either the inequality for x, y\ge 0 or change the two cases so that the equality goes with R_T^x<0.

- In the proof of Theorem 5.3, there is a line saying “from Lemma B.2 given in Appendix B.5 with a=…, b=…,” as the statement is not very long, I would personally prefer writing the statements out to make the body a bit more complete.

- Should also include \lambda in input for Alg 1.

**Strengths And Weaknesses:**

# Strengths

First of all, the paper introduces a seemingly interesting problem that could have good relevance in real-world scenarios. The implications of such a problem are also non-trivial, i.e., they show the lower bound, indicating an extra complexity by having contaminated functions. Also, they show the existing popular adaptive algorithms haven’t yet ready to achieve tight result. However, they go on to demonstrate that, with additional information, it is indeed possible to design an algorithm that meets the lower bound.

The paper is mostly well-written, with clear explanations and a logical flow, making it a pleasant read. Additionally, the work is quite comprehensive, offering a thorough analysis of several adaptive algorithms in this type of contaminated setting.

# Weakness

While the paper presents interesting results, there are several parts that could benefit from further clarification and improvement.

- Clarification on OGD in this setting: I might have overlooked, but I think it could be better to also give at least a few words about OGD in this setting.

- Corollary 5.4: In the proof of Corollary 5.4, the term $GDd\log T$ is discarded quite early. However, it seems necessary to address how $k$ compares with $\sqrt{d\log T}$, is it?

- Examples 3.7 and 3.8: The notation, functions, and the flow of the examples can be improved. For instance, the introduction of minibatch data and parameters feels a bit abrupt and unexplained. A smoother transition with clearer definitions and context for the parameters (e.g., minibatch size n) would make these examples more accessible.

- Lemma 6.1 and Theorem 6.2: The statements for these two results seem somewhat incomplete compared to other statements in the paper. In previous results, you provide a more detailed description of the function sequence. However, it is unclear how one would describe the specific scenario outlined in Section 6.

- Theorem 6.2 proof: The first inequality in the proof seems to move rather quickly. I could be missing something but not all functions involved are strongly convex, and it’s unclear how the definition of strong convexity is applied.

- Regret bound of ONS in A.2: Is the regret bound of ONS in A.2 $O((d/\alpha) \log T)$ or $O((d/\gamma) \log T)$?


## Regarding example 3.7, 3.8, Remark 3.9 and Appendix C.3

I decided to extract this part separately since I wasn't sure where it fit.

The implication of Remark 3.9 quite interesting, as it suggest a tradeoff regarding how small we can take parameters $\alpha$ or $\lambda$, and how many contaminated samples we can afford.

However, I wonder if you need to explicitly define $\lambda$ and $\alpha$ earlier before the algorithm starts, or if you can adapt in some way through learning. With the examples also illustrated in Appendix C.3, I’m still unclear about how these values were chosen or derived.

As for C.3, I also found some aspects of the results difficult to interpret. First, the overlap of the curves makes them hard to differentiate, especially in regions where they seem to almost coincide. Additionally, it looks like you used a relatively small T in the plot experiments - do you have any specific reason for focusing on such a small time horizon? Would a larger T provide more clarity in the results? You did seem to have a try on that though. Lastly, I noticed that Con-OGD seems to center quite closely with ONS. Do you have any insights or comments on why they appear this way?

---

> ### Author Response · Authors · 2024-10-10
>
> We appreciate your thorough review and valuable comments.
> We hope the following replies address your concerns and questions:
>
> ## Weaknesses
> > Clarification on OGD in this setting: I might have overlooked, but I think it could be better to also give at least a few words about OGD in this setting.
>
> When applying OGD with a $\Theta (1/t)$ stepsize to the instance described in Section 4.1, it incurs $\Omega(T)$ regret.
> Therefore, for $k$-contaminated strongly convex problems, OGD with this stepsize leads to $\Omega((1/\lambda)\log T+k)$ regret.
> However, it remains unclear whether OGD can achieve $O((1/\lambda)\log T+\sqrt{k})$ regret with a different stepsize.
> We have added Remark 4.7 in the manuscript (p.6).
>
> > Corollary 5.4: In the proof of Corollary 5.4, the term $GDd\log T$ is discarded quite early. However, it seems necessary to address how $k$ compares with $\sqrt{d\log T}$, is it?
>
> While the term is discarded, this is not due to $\sqrt{d\log T}=O(k)$ but because $GD=O(1/\gamma)$.
> We have clarified this in the proof (p.8).
>
> > Examples 3.7 and 3.8: The notation, functions, and the flow of the examples can be improved. For instance, the introduction of minibatch data and parameters feels a bit abrupt and unexplained. A smoother transition with clearer definitions and context for the parameters (e.g., minibatch size n) would make these examples more accessible.
>
> Thank you for the suggestion.
> We have added more detailed explanations to improve the clarity of the examples (p.4).
>
> > Lemma 6.1 and Theorem 6.2: The statements for these two results seem somewhat incomplete compared to other statements in the paper. In previous results, you provide a more detailed description of the function sequence. However, it is unclear how one would describe the specific scenario outlined in Section 6.
>
> Thank you for pointing this out.
> For clarity, we have added comments about $S_1$, $S_2$, $U$, and $k$ in the statements (p.9 and p.11).
>
> > Theorem 6.2 proof: The first inequality in the proof seems to move rather quickly. I could be missing something but not all functions involved are strongly convex, and it’s unclear how the definition of strong convexity is applied.
>
> Thank you for the observation.
> We have added further details to the inequality in the proof (p.11).
>
> > Regret bound of ONS in A.2: Is the regret bound of ONS in A.2 $O((d/\alpha) \log T)$ or $O((d/\gamma) \log T)$?
>
> Both $O((d/\alpha)\log T)$ and $O((d/\gamma)\log T)$ are correct.
> In fact, if $GD$ is taken as a constant, $\gamma=\alpha/2$ for sufficiently small $\alpha$.
> To maintain consistency with Table 1, we have revised the manuscript to use $O((d/\alpha)\log T)$ (p.14).
>
> ## Regarding examples 3.7, 3.8, Remark 3.9, and Appendix C.3
>
> > However, I wonder if you need to explicitly define $\lambda$ and $\alpha$ earlier before the algorithm starts, or if you can adapt in some way through learning. With the examples also illustrated in Appendix C.3, I’m still unclear about how these values were chosen or derived.
>
> In Experiment 3, we first set $k$ and compute the thresholds $\alpha$ and $\lambda$ based on $k$, although this is not always feasible in real applications.
> Since $k$ depends on $\alpha$ or $\lambda$, it is difficult to predetermine optimal values for these parameters.
> We have added this comment in Section C.3 (p. 29).
>
> > As for C.3, I also found some aspects of the results difficult to interpret. First, the overlap of the curves makes them hard to differentiate, especially in regions where they seem to almost coincide. Additionally, it looks like you used a relatively small T in the plot experiments - do you have any specific reason for focusing on such a small time horizon? Would a larger T provide more clarity in the results? You did seem to have a try on that though.
>
> The choice of small $T$ was due to computation time.
> However, recognizing the importance of this point, we conducted additional experiments using a larger $T$.
> The new figures show fewer overlaps, although some overlap still persists, and the results do not show a significant performance advantage of our proposed methods.
>
> > Lastly, I noticed that Con-OGD seems to center quite closely with ONS. Do you have any insights or comments on why they appear this way?
>
> We appreciate your observation, though we were unable to identify the specific point you referred to in this question.
>
> Additionally, our re-experiments revealed that the poor performance of OGD was due to the stepsize not being optimally tuned.
> OGD performs better when using the optimal stepsize, as shown in Experiment 3.

---

> ### Comment · Reviewer_vD1L · 2024-10-16
>
> * I edited it just to fix the format.
> ---
>
> Thanks for clarifying and adjusting the paper. Regarding my last question, "Con-OGD seems to center quite closely with ONS. Do you have any insights or comments on why they appear this way?" I was referring to Figure 3, but after increasing T, the question no longer exists.
>
> Thanks for the attempt for examples 3.7 and 3.8, but let me clarify a bit my suggestion. I was first more puzzled by where and how $(a_{t,i}, b_{t,i})$ comes and how they contribute to $f_t(x)$, and whether the "mini-batch" vs "online" mean anything particular in this regard. By having "mini-batch", I can imagine for the case in Example 3.7, at each time $t$, you receive/sample a mini-batch of n samples $\{(a_{t,i},b_{t,i})\}\_{i=1}^n$, which form the $f_t$ as you described, which is rather fine. For the case in Example 3.8, by using a different naming as "online" rather than "mini-batch" as in 3.7, I thought you might be only receiving a single pair of $(a_{t}, b_{t})$, and you are performing logistic regression on the increasing number of samples or whatever. However, the notations seem to suggest a similar scenario to Example 3.7, which confused me a bit. Therefore, I thought it could help for a smoother reading by specifying how you would imagine $\{(a_{t,i},b_{t,i})\}_{i=1}^n$ to come and name the examples more carefully.
>
> Apart from that, I don't have anything further to add.

---

> > ### Author Response · Authors · 2024-10-21
> >
> > We appreciate your suggestion for the examples.
> > Based on the feedback, we have modified the explanation of the examples to clarify that we are considering similar situations in the two examples.
> > We hope this revision addresses your concerns and enhances the clarity of our manuscript.

---

> > > ### Comment · Reviewer_vD1L · 2024-10-24
> > >
> > > I have no further comments and am looking forward to the work to be published.

---

### Decision · Action_Editor_6Bwu · 2024-10-26

**Recommendation:** Accept as is

**Comment:**

All three reviewers were happy with the paper already in its originally submitted form, requiring only small clarifications and adjustments. The authors have addressed these remarks satisfyingly in the updated version of the paper. Eventually, all reviewers recommended acceptance of the paper without changes. I concur with their assessment: this is a great paper that definitely deserves to be published at TMLR.

**Audience:**

The paper contributes to the abundant literature of online convex optimization, traditionally published at machine-learning (theory) venues like COLT, ALT, but also at NeurIPS and ICML, JMLR and more recently TMLR. Thus, a good portion of the journal's readership will find the results interesting.

**Claims And Evidence:**

The results of the paper are mainly theoretical, and all claims are supported with rigorous proofs.